# Contribution of New Three-Dimensional Code Based on the VWZCC Code Extension in Eliminating Multiple Access Interference in Optical CDMA Networks

**Mohamed Rahmani** [1], **Abdelhamid Cherifi** [2,*], **Abdullah S. Karar** [3], **Ghoutia Naima Sabri** [1,4] **and Boubakar S. Bouazza** [2]

1    The Information and Telecommunications Processing Laboratory (LTIT), Electrical Engineering Department, Faculty of Technology, TAHRI Mohamed University of Bechar, Bechar 08000, Algeria; rahmani.mohamed@univ-bechar.dz (M.R.); sabri.ghoutia@univ-bechar.dz (G.N.S.)
2    Laboratory of Technology of Communications (LTC), Electronics Department, Dr. Tahar Moulay University of Saida, Saida 20000, Algeria; bsbouazza@yahoo.fr
3    College of Engineering and Technology, American University of the Middle East, Kuwait; abdullah.karar@aum.edu.kw
4    Department of Material Sciences, Faculty of Exact Sciences, TAHRI Mohamed University of Bechar, Bechar 08000, Algeria
*    Correspondence: cherifi.abdelhamid@gmail.com or abdelhamid.cherifi@univ-saida.dz

**Abstract:** In order to solve the problem of one-dimensional code length, two-dimensional code spatial length, phase induced intensity noise PIIN effect, improved system capacity, and increased the number of simultaneous users, a new three-dimensional spectral/time/spatial variable weight zero cross correlation code for non-coherent spectral amplitude coding-optical code division multiple access (3D-VWZCC-SAC-OCDMA) is proposed in this paper. Its construction is based on a one-dimensional (1D) spectral sequence and two-dimensional (2D) temporal/spatial sequences, which are characterized by the property of zero cross correlation ZCC. The simulation results demonstrate that our code proves high immunity against PIIN noise and shot noise, it increases multiplexing ability when the passage is from (1D) and (2D) to (3D) up to 5.112 and 2.248 times, and it saves $-7.04$ dBm and $-5.9$ dBm of the receiver power due to simple detection at the receiver; furthermore, the 3D-VWZCC system capacity has outperformed the 3D-PD, 3D-PD/MD and 3D-DCS/MD codes, which reach 3686, 2908, and 3234 times, respectively. Moreover, our code offers better performance, in terms of data rates, with up to 2 Gbps compared to previous codes, which makes the system meet the requirements of optical communication networks. Further, 3D-VWZCC is also simulated in Optisystem software, where our code offers a transmission quality Q reaching 11.56 with a bit error rate BER of $1.99 \times 10^{-31}$ despite a high number of users.

**Keywords:** phase-induced intensity noise (PIIN); variable weight zero cross correlation (VWZCC) code; optical code division multiple access (OCDMA); bit error rate (BER)

## 1. Introduction

In previous years, communication networks depended on transmission through copper cables, but these cables could not cope with current technological developments due to several negative points provided by these cables, such as low transmission speed, significant attenuation, crosstalk, and short-distance transmission, which led to thinking in other solutions, including transmission by optical fibers [1,2].

Fiber optic-based transmission technology plays a big part in many fields thanks to several advantages, such as wide bandwidth, very high speed data, low attenuation, long distance transmission, and in addition to that, low price [3].

Recently, multiplexing techniques have become some of the essential technologies in the field of optical networks because of the efficiency they provide in terms of multiplexing

capacity and the transmitted data rate such as optical time division multiple access (OT-DMA), wavelength division multiple access (WDMA), and optical code division multiple access (OCDMA) [4–6]. In optical time division multiple access and wavelength division multiple access, each user uses a time slot and a wavelength to access and share the data in the channel, respectively. As a result, an increase in the number of users causes an increase in temporal duration and optical band. Therefore, this increase is directly linked to the increase in the number of subscribers, which leads to degradation in the performance of the system [6,7].

Optical code division multiple access (OCDMA) multiplexing is one of the techniques that improves the quality of service (QOS) and increases the capacity of the system, where each user uses a different optical signature than the others to send their information to the receiver through an optical channel synchronously and asynchronously [5,8,9], and all the bandwidth of the channel is used for an indefinite period of time. Thanks to these characteristics, this technique guarantees transmission security, data security, and transparency between users during communication [10,11]. Furthermore, the OCDMA network is divided into two categories according to the nature of the optical source: coherent OCDMA and incoherent OCDMA. In coherent OCDMA, the generated optical codes are composed of (−1) and (+1); therefore, we say that these codes are strictly orthogonal and bipolar. Moreover, the system becomes more complex and very expensive due to the use of these bipolar codes. Incoherent OCDMA optical codes are generated using binary data (0) and (+1). As a result, the codes are not strictly orthogonal and unipolar; for that, the system requires very simple optical devices and is less expensive [12].

Generally, most OCDMA optical code multiplexing systems are based on transmission through an incoherent optical source due to low transmission power and wide bandwidth. However, this technique is characterized by multiple access interference (MAI), which limits the ability to produce the maximum number of optical codes due to the property of cross-correlation of the codes, which are almost orthogonal depending on the construction method and create phase-induced intensity noise (PIIN), which causes multi-user interference (MUI) and leads to influence on the system performance [10,11,13,14].

To eliminate these disadvantages, the SAC-OCDMA system is considered an important key to reducing multiple access interference MAI, which is defined by the intersection of the chips of the ones (1) [15–18], and improves system performance through high immunity against localized noise, very low overall power spectral density DSP, and the encoding and decoding devices, which are simple and less expensive. The development of this field has led to the creation of several codes of different dimensions, according to the various domains: temporal, spatial, polarization, or spectral. The first approach is the one-dimensional codes, which are carried out in time or in the spectrum. Further, there are four types of encoding: spread spectrum by direct sequence OCDMA (DS-SS-OCDMA), which is based on the concept of binary sequence spreading of each user; secondly, the temporal phase encoding (TPE: temporal phase-encoded OCDMA) where the data is modulated by the phase; thirdly, the spectral encoding in amplitude (SAC: spectral amplitude coding OCDMA), which consists of modulating the spectrum of each user by an amplitude value; fourthly, the spectral encoding in phase (SPE: spectral phase encoding OCDMA), where the spectrum is modulated by phase modulation [2,4,11,13].

In the 1D case, two families of codes exist according to the construction method: the codes with non-zero cross-correlation and the codes with zero cross-correlation. The first family is characterized by the presence of PIIN noise such as FCC, MDW, DEU, and DCS codes published in [19–22], whereas the second family is characterized by the absence of this noise such as ZCC, PTZCC, and MD, published in [5,8,23,24]. Moreover, the optical code length is directly linked by the increase in the number of users; this property causes a significant degradation in terms of multiplexing capacity and the signal-to-noise ratio SNR.

In order to solve the problem of code length and improve the capacity of the one-dimensional system, studies have been directed towards two-dimensional codes; therefore, several researchers have proposed two-dimensional codes by combining between two

parameters among these parameters: spectral, spatial, time, and polarization. Currently, there are three combinations (spectral/time, spectral/spatial, and time/spatial). In [5], the Single Weight Zero Cross-Correlation code based on 1D-SWZCC was proposed, and furthermore, the PIIN noise is completely suppressed due to the property of zero cross-correlation. In [7], a two-dimensional half spectral/spatial zero cross-correlation code has been proposed, where the effect of PIIN is always absent; in addition, the system complexity has been reduced to half thanks to the ZCC property. In [8], Pascal's triangle zero cross-correlation code is developed. In [25], the Perfect Difference code is constructed to reduce the effect of PIIN. In [26], the two-dimensional Dynamic Cyclic Shift code has been developed for the purpose of removing PIIN noise by using a method of cancellation property of the MAI. In [27], two-dimensional Diagonal Eigen value Unity code is also proposed. In [28], the two-dimensional multi-diagonal code is proposed.

Despite the advantages of two-dimensional codes, the increase in the capacity of a two-dimensional system makes it possible to increase the spatial or even temporal length depending on the code construction method. This is due to the relation that links the spatial or temporal length to the number of subscribers. Thus, an increase in the number of users causes an increase in the number of spatial components or the slot duration (time difference) of each user at spectral/spatial and spectral/time, respectively. This increase leads to the degradation of performance and increases the error rate; the system becomes very complex due to a large number of encoding and decoding devices. To control these negative characteristics, several codes have been realized in order to reduce the code length and make the system applicable. In [10], the three-dimensional code (3D-SWZCC) is also developed to eliminate the effect of PIIN based on the two-dimensional spectral/time code 2D-SWZCC, and a one sequence 1D-SWZCC for the spatial code, where the system became efficient due to the ZCC property. In [11], a three-dimensional code (3D-PTZCC) is based on three sequences of the same PTZCC code, where PIIN is suppressed due to the property of ZCC. In [12], a three-dimensional multi-diagonal (3D-MD) code has been developed where the PIIN noise is totally suppressed. In [13], a three-dimensional perfect difference code (3D-PD) code is proposed in order to remove the effect of PIIN. In [14], a three-dimensional hybrid code is called the perfect difference/multi-diagonal (3D-PD/MD) based on two sequences and one sequence of 2D-PD and 1D-MD code, respectively.

Accordingly, this study proposes a new spectral/time/spatial code called the variable weight zero cross correlation for non-coherent OCDMA (3D-VWZCC-OCDMA) system, which is characterized by the property of zero cross correlation, simple construction, simple detection, supports large number of users, eliminates the effects of PIIN noise, and has low power consumption. As a result, the suggested code gives better performance in terms of bit error rate and signal-to-noise ratio at a higher data rate.

In this context, the rest of this paper is composed as follows: Section 2 presents a literature review of some previous codes. Section 3 describes the method of code construction. The presentation and description for a non-coherent system based on the new code is in Section 4. Then, Section 5 introduces the mathematical formulas of the three-dimension code (3D-VWZCC). The numerical results and the simulation of the system using MATLAB and Optisystem, respectively, are presented in Sections 6 and 7, respectively. Finally, we end with a conclusion about the code and its performance in Section 8.

## 2. Literature Review

The literature review of the various codes developed and studied previously is presented in Table 1.

**Table 1.** Literature review of the various codes.

| Codes | Benefits | Drawbacks |
|---|---|---|
| **1D-FCC** | - Short Code Length | - AND Subtraction (Two Photo-detectors)<br>- Non-Zero Cross Correlation $\lambda_c \geq 1$<br>- Code Weight $W \geq 2$ (Even Number)<br>- Support Low Number of Users<br>- Low Data Rate |
| **1D-MDW** | - Short Code Length | - AND Subtraction (Two Photo-detectors)<br>- Non-Zero Cross Correlation $\lambda_c = 1$<br>- Code Weight $W > 2$<br>- Support Low Number of Users<br>- Low Data Rate |
| **1D-DCS** | - Short Code Length | - AND Subtraction (Two Photo-detectors)<br>- Non-Zero Cross Correlation $\lambda_c = 1$<br>- Code Weight $W \geq 1$ (Any Number)<br>- Support Low Number of Users<br>- Low Data Rate |
| **1D-DEU** | - Short Code Length | - XOR Subtraction Detection Technique<br>- Non Zero Cross Correlation $\lambda_c \leq 1$<br>- Code Weight $W \geq 1$ (Any Number)<br>- Support Low Number of Users<br>- Low Data Rate |
| **1D-MD** | - Support Large Number of Users<br>- High Data Rate<br>- Zero Cross Correlation<br>- PIIN = 0<br>- Absence Multi Access Interferences (MAI)<br>- Direct Detection Technique<br>- Zero Cross Correlation $\lambda_c = 0$<br>- Code Weight $W \geq 1$ (Any Number) | - Long Code Length<br>- Needs Narrow Filters |
| **1D-ZCC** | - Support Large Number of Users<br>- High Data Rate<br>- Zero Cross Correlation<br>- PIIN = 0<br>- Absence Multi Access Interferences (MAI)<br>- Direct Detection Technique<br>- Zero Cross Correlation $\lambda_c = 0$<br>- Code Weight $W \geq 1$ (Any Number) | - Compatible with Even Number of Users<br>- Long Code Length |
| **1D-PTZCC** | - Support Large Number of Users<br>- High Data Rate<br>- Zero Cross Correlation<br>- PIIN = 0<br>- Absence Multi Access Interferences (MAI)<br>- Direct Detection Technique<br>- Zero Cross Correlation $\lambda_c = 0$<br>- Code Weight $W \geq 1$ (Any Number) | - Long Code Length |
| **2D-PD** | - High Data Rate<br>- Short Code Length Compared To 1D | - High PIIN Noise<br>- Presence MAI<br>- Two Photo-detectors at The Receiver |
| **2D-DCS** | - High Data Rate<br>- Short Code Length Compared To 1D | - High PIIN Noise<br>- Presence MAI<br>- Two Photo-detectors at The Receiver |

**Table 1.** *Cont.*

| Codes | Benefits | Drawbacks |
|---|---|---|
| **2D-DEU** | - High Data Rate<br>- Short Code Length Compared To 1D | - High PIIN Noise<br>- Presence MAI<br>- Two Photo-detectors at The Receiver |
| **2D-MD** | - High Data Rate<br>- Short Code Length Compared To 1D<br>- Supports High Number of Users | - Presence MAI<br>- Needs Narrow Filters |
| **2D-SWZCC** | - High Data Rate<br>- Short Code Length Compared To 1D<br>- Supports High Number of Users<br>- Low PIIN Noise | - Single Weight W = 1 |
| **2D-PTZCC** | - High Data Rate<br>- Short Code Length<br>- Supports High Number of Users<br>- Low PIIN Noise | - Needs Narrow Filters |
| **2D-HSSZCC** | - High Data Rate<br>- Low PIIN Noise<br>- Short Code Length Compared To 1D and 2D<br>- Supports High Number of Users | - Needs Narrow Filters<br>- Compatible with Even Number of Users |
| **3D-PD** | - High Data Rate<br>- Short Code Length Compared To 1D and 2D | - Two Photo-detectors at The Receiver<br>- More Complex<br>- Higher Power Consumption |
| **3D-PD/MD** | - High Data Rate<br>- Short Code Length | - Two Photo-detectors at The Receiver<br>- More Complex |
| **3D-MD** | - High Data Rate<br>- Short Code Length Compared To 1D And 2D<br>- Supports High Number of Users<br>- Direct Detection Technique | - Needs Narrow Filters |
| **3D-SWZCC** | - High Data Rate<br>- Short Code Length Compared To 1D and 2D<br>- Supports High Number of Users<br>- Direct Detection Technique | - Single Weight W = 1 |
| **3D-PTZCC** | - High Data Rate<br>- Short Code Length Compared To 1D and 2D<br>- Supports High Number of Users<br>- Direct Detection Technique | - Needs Narrow Filters |
| **Proposed 3D-VWZCC** | - High Data Rate<br>- Short Code Length Compared To 1DAnd 2D<br>- Supports High Number of Users<br>- Direct Detection Technique<br>- Simple Architecture<br>- Absence MAI<br>- Very Low PIIN Noise<br>- Admits Any Number of Users and Any Code Weight<br>- Needs Simple Filters | - No Drawbacks |

In [19], a new design for spectral coding, called the Flexible Cross-Correlation Code (FCC), has been proposed. The suggested code has advantages such as flexible cross-correlation property for any number of users and any weight; furthermore, the proposed code provides better performance in terms of the bit error rate (BER) compared to the Modified Double Weight (MDW), dynamic cyclic shift (DCS), Modified Frequency Hopping (MFH) and Hadamard codes. Despite this, this code is characterized by PIIN noise due to the intersections between the code spectral chips, which degrade the performance of the optical system.

In [20], a new design coding for OCDMA systems called the Modified Double Weight (MDW) code has been developed for the non-coherent OCDMA system. Therefore, the developed code gives better performance compared to the modified frequency hopping (MFH) and Hadamard codes in terms of BER for a large number of users. However, the code is also characterized by PIIN noise, which degrades the system performance as the number of users increases.

In [21], a new one-dimensional spectral coding family called the Diagonal Eigenvalue Unity (DEU) code has been developed, where the proposed code is compatible with any weight; moreover, the code performance demonstrates that the DEU code gives better performance compared to random diagonal (RD), DCS, MFH, and modified quadratic congruence (MQC) codes in terms of the signal to noise ratio (SNR) and BER. Despite this, the increase in the number of users creates multiple access interference (MAI) thanks to a non-zero cross-correlation property, which leads to limiting the number of simultaneous users.

In [22], a new code named the dynamic cyclic shift (DCS) has been suggested for optical CDMA systems. Further, the code supported a large number of users and any code weight; furthermore, the system performance demonstrates that the DCS code provides better performance in terms of BER as a function of the number of users at a high data rate.

In [23], a zero cross-correlation code called multi-diagonal for one-dimensional spectral coding has been suggested. Moreover, the code is compared with MQC and RD codes; as a result, the results demonstrate that the MD code provides better performance in terms of SNR and BER as a function of the number of simultaneous users due to the ZCC property.

In [8], a new zero cross-correlation code called Pascal's triangle zero cross-correlation 1D-PTZCC for OCDMA system has been studied, where the proposed code totally suppressed the PIIN noise thanks to the property of ZCC.

In [7,24], a new zero cross-correlation code 1D-ZCC for the OCDMA-orthogonal frequency division multiplexing (OFDM) system has been studied. Furthermore, the code supports a large number of users and offers better performance due to the property of zero cross-correlation.

In [5], a single-weight zero cross-correlation (2D-SWZCC) code based on 1D-SWZCC has been proposed; furthermore, the PIIN noise is completely suppressed due to zero cross-correlation properties. Further, the code supports a large number of users at a high data rate; despite this, this code is used in the case where the code weight is unique.

In [7], a two-dimensional half spectral/spatial zero cross-correlation (2D-HSSZCC) code has been proposed, where the effect of PIIN is always absent; in addition, the system complexity has been reduced to half thanks to the ZCC property. Despite this, the code is compatible with an even number of users and needs narrow filters at the receiver level.

In [8], Pascal's triangle zero cross-correlation (2D-PTZCC) code is developed where the code supports a large number of users, needs narrow filters, and is affected by a low PIIN noise.

In [26], a two-dimensional dynamic cyclic shift (2D-DCS) code is developed for the purpose of removing PIIN noise by using a method of cancellation property of the MAI. The code is characterized by high PIIN noise, which leads to the presence of MAI, and needs two photo-detectors at the receiver to detect the optical signal.

In [27], a two-dimensional Diagonal Eigen value Unity (2D-DEU) code is also proposed. Furthermore, the 2D-DEU code is affected by a high PIIN noise due to non-zero cross-correlation, which leads to multi-access interference MAI.

In [28], the two-dimensional multi-diagonal (2D-MD) code is proposed in order to reduce PIIN noise. As a result, the code is characterized by MAI and needs narrow filters to detect an optical signal at the receiver.

In [10], the three-dimensional (3D-SWZCC) code is also developed to eliminate the effect of PIIN based on the two-dimensional spectral/time code 2D-SWZCC, and a one sequence 1D-SWZCC for the spatial code, where the system becomes efficient due to the ZCC property and supports a large number of users. Despite that, the code is still only compatible with a single weight.

In [11], the three-dimensional code (3D-PTZCC) based on three sequences of the same PTZCC code where PIIN is suppressed due to the property of ZCC, and the optical signal is detected by the direct detection technique. As well, the detection at the receiver needs narrow filters.

In [12], the three-dimensional multi-diagonal (3D-MD) code has been developed where the PIIN noise is totally suppressed. Moreover, the code supports a high data rate compared to the 1D and 2D codes, and the detection of the optical signal needs a narrow filter.

In [13], the three-dimensional perfect difference code (3D-PD) code is proposed in order to remove the effect of PIIN, where this code needs two photo-detectors in order to detect the optical signal at the receiver, which allows the system becomes more complex.

In [14], the three-dimensional hybrid code called the perfect difference/multi-diagonal (3D-PD/MD) based on two sequences and one sequence of 2D-PD and 1D-MD code, respectively, has been studied. The suggested code also needs two photo-detectors at the receiver, which allows the system becomes more complex and needs higher power consumption.

## 3. 3D-VWZCC Code Construction

The three-dimensional code, 3D-VWZCC, is built from two sequences of one-dimensional and two-dimensional 1D-VWZCC, 2D-VWZCC codes, respectively. To construct this code, there are three essential parameters, which are presented as follows: code length ($L$), code weight ($W$), and cross-correlation value ($\lambda_c$). The 1D-VWZCC code is characterized by the property of zero cross-correlation ZCC; as a result, the code length and the value of cross-correlation are given by the following equations:

$$\begin{cases} \lambda_c = 0 \\ L = K \times W \end{cases} \tag{1}$$

where $K$ is the number of users; further, due to the ZCC property the PIIN being totally suppressed, the construction is described by four steps:

### 3.1. 1D-VWZCC Code Construction

*Step 1*

First, we generate a vector $V_c$ depending on the weight value and number of users, where the number of zeros is $K$-$W$:

$$V_c = \begin{bmatrix} 1 \\ \vdots \\ 1 \\ 0 \\ \vdots \\ \vdots \\ \vdots \\ 0 \end{bmatrix} \tag{2}$$

*Step 2*

A single-bit vertical shift operation is applied to the vector $V_c$ where the number of shift operations is equal to $N_d = K - 1$

For example, $K = 3$, $W = 2$, the number of shift operations equal to $N_d = 2$, and we obtain three vectors $V_c$

$$V_{C1} = \begin{bmatrix} 1 \\ 1 \\ 0 \end{bmatrix}, \ V_{C2} = \begin{bmatrix} 0 \\ 1 \\ 1 \end{bmatrix}, \ V_{C3} = \begin{bmatrix} 1 \\ 0 \\ 1 \end{bmatrix} \tag{3}$$

*Step 3*

Novel matrix [P] designed from vectors $V_{cs}$, is a combination between vectors $V_{cs}$:

$$P = [V_{ci}] = \begin{bmatrix} V_{c1} & V_{c2} & \ldots & \ldots & V_{ck} \end{bmatrix} \tag{4}$$

where $i = 1, \ldots, K$

$$P = [V_{ci\ldots\ldots K}] = \begin{bmatrix} V_{c1} & V_{c2} & V_{c3} \end{bmatrix} = \begin{bmatrix} \begin{bmatrix} 1 \\ 1 \\ 0 \end{bmatrix} & \begin{bmatrix} 0 \\ 1 \\ 1 \end{bmatrix} & \begin{bmatrix} 1 \\ 0 \\ 1 \end{bmatrix} \end{bmatrix}$$

Hence, *P* can be written as:

$$P = \begin{bmatrix} 1 & 0 & 1 \\ 1 & 1 & 0 \\ 0 & 1 & 1 \end{bmatrix} \tag{5}$$

After that, we notice there are overlaps between the ones (1); to remove these interferences, a new horizontal summation method '*H*' has been used, as defined by the following relation:

$$H = \sum_{i=1}^{C} C_i \tag{6}$$

where *H* is the sum of the bits of each user, $C_i$ is a binary bit (0 or 1), and *C* is the length of code.

We take the previous matrix *P*:

$$H_1 = \sum_{i=1}^{3} C_i = 1 + 0 + 1 = 2, \ H_2 = \sum_{i=1}^{3} C_i = 1 + 1 + 0 = 2, \ H_3 = \sum_{i=1}^{3} C_i = 0 + 1 + 1 = 2$$

$$P = \begin{bmatrix} 1 & 0 & 1 \\ 1 & 1 & 0 \\ 0 & 1 & 1 \end{bmatrix} \begin{matrix} \rightarrow & H_1 \\ \rightarrow & H_2 \\ \rightarrow & H_3 \end{matrix}$$

The value of *H* is equal to 2 for all codes and is the same value of weight *W*, so we can write *H* as *W*:

$$P = \begin{bmatrix} \ldots & H & \ldots \\ \ldots & H & \ldots \\ \ldots & H & \ldots \end{bmatrix} = \begin{bmatrix} \ldots & W & \ldots \\ \ldots & W & \ldots \\ \ldots & W & \ldots \end{bmatrix}$$

The positions of the Hs in the *P* matrix are diagonal and the rest is filled with zeros. Accordingly, the result of the matrix *P* is divided on *H* in order to obtain the identity matrix of size $K \times K$ denoted by $P_I$:

$$P_I = \begin{bmatrix} P \\ \overline{H} \end{bmatrix} = \cfrac{\begin{bmatrix} H & 0 & 0 \\ 0 & H & 0 \\ 0 & 0 & H \end{bmatrix}}{H} = \begin{bmatrix} \frac{H}{H} & 0 & 0 \\ 0 & \frac{H}{H} & 0 \\ 0 & 0 & \frac{H}{H} \end{bmatrix} \tag{7}$$

For the previous example:

$$P_I = \left[\frac{P}{H}\right] = \left[\frac{\begin{array}{ccc} 3 & 0 & 0 \\ 0 & 3 & 0 \\ 0 & 0 & 3 \end{array}}{3}\right] = \begin{bmatrix} \frac{3}{3} & 0 & 0 \\ 0 & \frac{3}{3} & 0 \\ 0 & 0 & \frac{3}{3} \end{bmatrix} = \begin{bmatrix} 1 & 0 & 0 \\ 0 & 1 & 0 \\ 0 & 0 & 1 \end{bmatrix}_{K \times K}$$

*Step 4*

Depending on the parameters $K$ and $W$ in the first phase, the matrix $P_I$ has a weight equal to 1 for each code, whereas the weight must be equal to 2. For that, the sub-columns $A_i$ of matrix $P_I$ are regenerated several times according to the requested weight.

$$P_I = [A_i] = \begin{bmatrix} A_1 & A_2 & \dots & \dots & A_k \end{bmatrix} \tag{8}$$

$$P_I = \begin{bmatrix} \begin{bmatrix} 1 \\ 0 \\ 0 \end{bmatrix} & \begin{bmatrix} 0 \\ 1 \\ 0 \end{bmatrix} & \begin{bmatrix} 0 \\ 0 \\ 1 \end{bmatrix} \end{bmatrix}_{K \times K} \tag{9}$$

$$A_1 = \begin{bmatrix} 1 \\ 0 \\ 0 \end{bmatrix}, \ A_2 = \begin{bmatrix} 0 \\ 1 \\ 0 \end{bmatrix}, \ A_3 = \begin{bmatrix} 0 \\ 0 \\ 1 \end{bmatrix}$$

Sub-matrices denoted by $X_i$ are obtained and expressed as:

$$X_i = [(W)times[A_i]] \tag{10}$$

where $i = 1, \dots, K$

$$X_1 = (W)times[A_1] = \begin{bmatrix} 1 & 1 \\ 0 & 0 \\ 0 & 0 \end{bmatrix}, X_2 = (W)times[A_2] = \begin{bmatrix} 0 & 0 \\ 1 & 1 \\ 0 & 0 \end{bmatrix}, \ X_3 = (W)times[A_3] = \begin{bmatrix} 0 & 0 \\ 0 & 0 \\ 1 & 1 \end{bmatrix}$$

For the last time, the one-dimensional code called the variable weight zero cross correlation VWZCC is generated by merging the sub-matrices $X_i$. An example of 1D-VWZCC is shown in Table 2, when $K = 5$, $W = 2$

$$1D - VWZCC = [X_i] = [X_1 \dots\dots\dots X_K] \tag{11}$$

$$1D - VWZCC = [X_i] = [(W)times[A_i]] = \begin{bmatrix} (W)times[A_1] & \dots & \dots & \dots & (W)times[A_K] \end{bmatrix}$$

**Table 2.** 1D-VWZCC when $K = 5$ and $W = 2$.

| $VWZCC =$ | 1 | 1 | 0 | 0 | 0 | 0 | 0 | 0 | 0 | 0 |
|-----------|---|---|---|---|---|---|---|---|---|---|
|           | 0 | 0 | 1 | 1 | 0 | 0 | 0 | 0 | 0 | 0 |
|           | 0 | 0 | 0 | 0 | 1 | 1 | 0 | 0 | 0 | 0 |
|           | 0 | 0 | 0 | 0 | 0 | 0 | 1 | 1 | 0 | 0 |
|           | 0 | 0 | 0 | 0 | 0 | 0 | 0 | 0 | 1 | 1 |

### 3.2. 3D-VWZCC Code Construction

Let X, Y, and Z be three spectral, time, spatial code sequences, with $M$, $N$, $P$, which are the code lengths associated by each sequence, respectively, and given as $M = K_1 \times W_1$, $N = K_2 \times W_2$, $P = K_3 \times W_3$, where the combination $(K_i, W_i)$ is a number of users and code

weight of each component. An example of the 3D-VWZCC code is shown in Table 3, when $K_1 = 2$, $K_2 = K_3 = 3$

The relation of building the three-dimensional code is expressed by the following Equation [10–13]:

$$A_{g,h,l} = X_g^T Y_h Z_l \tag{12}$$

where $X_g^T$ is the $g^{th}$ sequence code of X, $Y_h$ is the $h^{th}$ sequence code of Y, and $Z_l$ is $l^{th}$ sequence code of Z, where $g = 1, 2, 3, \ldots, K_1$, $h = 1, 2, 3, \ldots, K_2$ and $l = 1, 2, 3, \ldots, K_3$.

The total capacity of the 3D-VWZCC code is expressed by the relation below; it depends on the number of users of X, Y, and Z [10–12]:

$$K = K_1 \times K_2 \times K_3 \tag{13}$$

To clarify the 3D-VWZCC code property, the characteristic matrices $A^{(d)} = A^{(0)}$, $A^{(1)}, \ldots A^{(7)}$ are given as [13]:

$$A^{(d)} = \begin{cases} \begin{array}{l} A^{(0)} = X^T Y Z \\ A^{(1)} = X^T \overline{Y} Z \\ A^{(2)} = \overline{X}^T Y Z \\ A^{(3)} = \overline{X}^T \overline{Y} Z \end{array} \quad \begin{cases} A^{(4)} = X^T Y \overline{Z} \\ A^{(5)} = X^T \overline{Y} \overline{Z} \\ A^{(6)} = \overline{X}^T Y \overline{Z} \\ A^{(7)} = \overline{X}^T \overline{Y} \overline{Z} \end{cases} \end{cases} \tag{14}$$

$X^T$ is the transpose of the matrix X. $\overline{X}$, $\overline{Y}$, and $\overline{Z}$ are the complements of the sequences codes of the matrices X, Y, and Z, respectively.

An example of the 3D-VWZCC code is represented in Figure 1, and it presents a three-dimensional graphical presentation according to the first code of VWZCC indicated in Table 3 where the weight of the spectral, temporal, and spatial sequences are, respectively, equal to $W_1 = W_2 = W_3 = 2$.

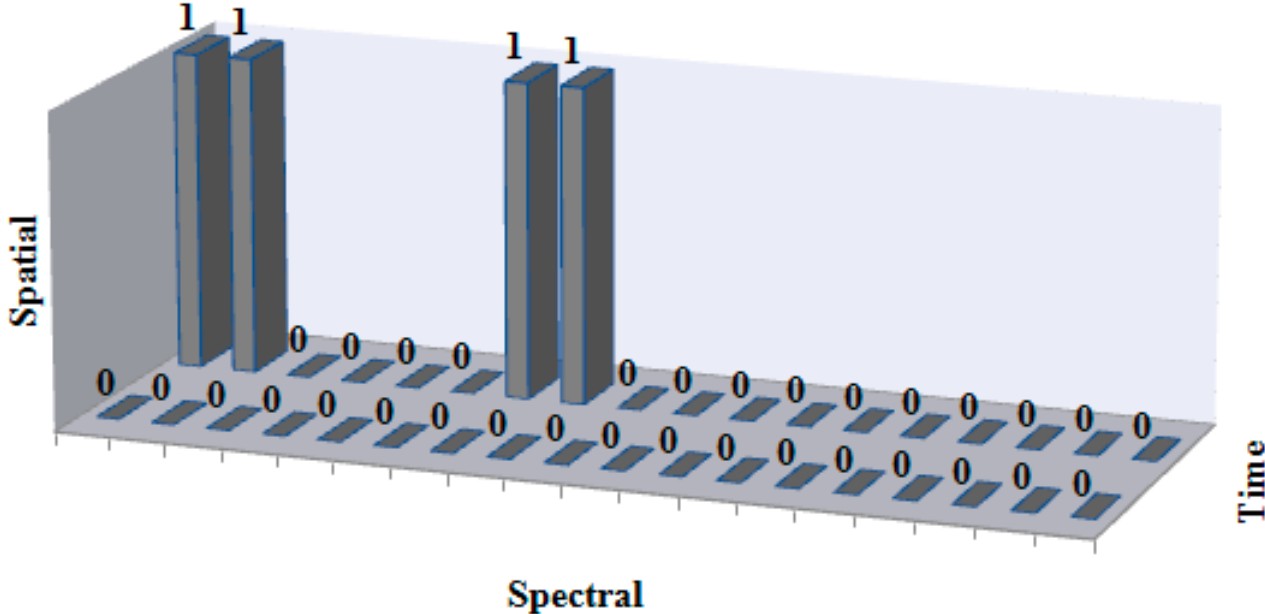

**Figure 1.** The 3D-VWZCC (spectral/time/spatial) code for the first users.

**Table 3.** The 3D-VWZCC code for $K_1 = 2$, $K_2 = K_3 = 3$.

$Y_0$=[ 1　1　0　0　0　0 ] , $Z_0$=[ 1　1　0　0　0　0 ]

$X_0 = \begin{bmatrix} 1 \\ 0 \end{bmatrix}$　$X_1 = \begin{bmatrix} 0 \\ 1 \end{bmatrix}$

$\begin{bmatrix} 1 & 1 & 0 & 0 & 0 & 0 & & 1 & 1 & 0 & 0 & 0 & 0 & & 0 & 0 & 0 & 0 & 0 & 0 \\ 0 & 0 & 0 & 0 & 0 & 0 & & 0 & 0 & 0 & 0 & 0 & 0 & & 0 & 0 & 0 & 0 & 0 & 0 \\ 0 & 0 & 0 & 0 & 0 & 0 & & 0 & 0 & 0 & 0 & 0 & 0 & & 0 & 0 & 0 & 0 & 0 & 0 \\ 1 & 1 & 0 & 0 & 0 & 0 & & 1 & 1 & 0 & 0 & 0 & 0 & & 0 & 0 & 0 & 0 & 0 & 0 \end{bmatrix}$

$Y_1$=[ 0　0　1　1　0　0 ] , $Z_0$=[ 1　1　0　0　0　0 ]

$X_0 = \begin{bmatrix} 1 \\ 0 \end{bmatrix}$　$X_1 = \begin{bmatrix} 0 \\ 1 \end{bmatrix}$

$\begin{bmatrix} 0 & 0 & 1 & 1 & 0 & 0 & & 1 & 1 & 0 & 0 & 0 & 0 & & 0 & 0 & 0 & 0 & 0 & 0 \\ 0 & 0 & 0 & 0 & 0 & 0 & & 0 & 0 & 0 & 0 & 0 & 0 & & 0 & 0 & 0 & 0 & 0 & 0 \\ 0 & 0 & 0 & 0 & 0 & 0 & & 0 & 0 & 0 & 0 & 0 & 0 & & 0 & 0 & 0 & 0 & 0 & 0 \\ 0 & 0 & 1 & 1 & 0 & 0 & & 1 & 1 & 0 & 0 & 0 & 0 & & 0 & 0 & 0 & 0 & 0 & 0 \end{bmatrix}$

$Y_2$=[ 0　0　0　0　1　1 ] , $Z_0$=[ 1　1　0　0　0　0 ]

$X_0 = \begin{bmatrix} 1 \\ 0 \end{bmatrix}$　$X_1 = \begin{bmatrix} 0 \\ 1 \end{bmatrix}$

$\begin{bmatrix} 0 & 0 & 0 & 0 & 1 & 1 & & 1 & 1 & 0 & 0 & 0 & 0 & & 0 & 0 & 0 & 0 & 0 & 0 \\ 0 & 0 & 0 & 0 & 0 & 0 & & 0 & 0 & 0 & 0 & 0 & 0 & & 0 & 0 & 0 & 0 & 0 & 0 \\ 0 & 0 & 0 & 0 & 0 & 0 & & 0 & 0 & 0 & 0 & 0 & 0 & & 0 & 0 & 0 & 0 & 0 & 0 \\ 0 & 0 & 0 & 0 & 1 & 1 & & 1 & 1 & 0 & 0 & 0 & 0 & & 0 & 0 & 0 & 0 & 0 & 0 \end{bmatrix}$

$Y_0$=[ 1　1　0　0　0　0 ] , $Z_1$=[ 0　0　1　1　0　0 ]

$X_0 = \begin{bmatrix} 1 \\ 0 \end{bmatrix}$　$X_1 = \begin{bmatrix} 0 \\ 1 \end{bmatrix}$

$\begin{bmatrix} 0 & 0 & 0 & 0 & 0 & 0 & & 1 & 1 & 0 & 0 & 0 & 0 & & 0 & 0 & 1 & 1 & 0 & 0 \\ 0 & 0 & 0 & 0 & 0 & 0 & & 0 & 0 & 0 & 0 & 0 & 0 & & 0 & 0 & 0 & 0 & 0 & 0 \\ 0 & 0 & 0 & 0 & 0 & 0 & & 0 & 0 & 0 & 0 & 0 & 0 & & 0 & 0 & 0 & 0 & 0 & 0 \\ 0 & 0 & 0 & 0 & 0 & 0 & & 1 & 1 & 0 & 0 & 0 & 0 & & 0 & 0 & 1 & 1 & 0 & 0 \end{bmatrix}$

$Y_1$=[ 0　0　1　1　0　0 ] , $Z_1$=[ 0　0　1　1　0　0 ]

$X_0 = \begin{bmatrix} 1 \\ 0 \end{bmatrix}$　$X_1 = \begin{bmatrix} 0 \\ 1 \end{bmatrix}$

$\begin{bmatrix} 0 & 0 & 0 & 0 & 0 & 0 & & 0 & 0 & 1 & 1 & 0 & 0 & & 0 & 0 & 1 & 1 & 0 & 0 \\ 0 & 0 & 0 & 0 & 0 & 0 & & 0 & 0 & 0 & 0 & 0 & 0 & & 0 & 0 & 0 & 0 & 0 & 0 \\ 0 & 0 & 0 & 0 & 0 & 0 & & 0 & 0 & 0 & 0 & 0 & 0 & & 0 & 0 & 0 & 0 & 0 & 0 \\ 0 & 0 & 0 & 0 & 0 & 0 & & 0 & 0 & 1 & 1 & 0 & 0 & & 0 & 0 & 1 & 1 & 0 & 0 \end{bmatrix}$

$Y_2$=[ 0　0　0　0　1　1 ] , $Z_1$=[ 0　0　1　1　0　0 ]

$X_0 = \begin{bmatrix} 1 \\ 0 \end{bmatrix}$　$X_1 = \begin{bmatrix} 0 \\ 1 \end{bmatrix}$

$\begin{bmatrix} 0 & 0 & 0 & 0 & 0 & 0 & & 0 & 0 & 0 & 0 & 1 & 1 & & 0 & 0 & 1 & 1 & 0 & 0 \\ 0 & 0 & 0 & 0 & 0 & 0 & & 0 & 0 & 0 & 0 & 0 & 0 & & 0 & 0 & 0 & 0 & 0 & 0 \\ 0 & 0 & 0 & 0 & 0 & 0 & & 0 & 0 & 0 & 0 & 0 & 0 & & 0 & 0 & 0 & 0 & 0 & 0 \\ 0 & 0 & 0 & 0 & 0 & 0 & & 0 & 0 & 0 & 0 & 1 & 1 & & 0 & 0 & 1 & 1 & 0 & 0 \end{bmatrix}$

$Y_0$=[ 1　1　0　0　0　0 ] , $Z_2$=[ 0　0　0　0　1　1 ]

$X_0 = \begin{bmatrix} 1 \\ 0 \end{bmatrix}$　$X_1 = \begin{bmatrix} 0 \\ 1 \end{bmatrix}$

$\begin{bmatrix} 1 & 1 & 0 & 0 & 0 & 0 & & 0 & 0 & 0 & 0 & 0 & 0 & & 0 & 0 & 0 & 0 & 1 & 1 \\ 0 & 0 & 0 & 0 & 0 & 0 & & 0 & 0 & 0 & 0 & 0 & 0 & & 0 & 0 & 0 & 0 & 0 & 0 \\ 0 & 0 & 0 & 0 & 0 & 0 & & 0 & 0 & 0 & 0 & 0 & 0 & & 0 & 0 & 0 & 0 & 0 & 0 \\ 1 & 1 & 0 & 0 & 0 & 0 & & 0 & 0 & 0 & 0 & 0 & 0 & & 0 & 0 & 0 & 0 & 1 & 1 \end{bmatrix}$

$Y_1$=[ 0　0　1　1　0　0 ] , $Z_2$=[ 0　0　0　0　1　1 ]

$X_0 = \begin{bmatrix} 1 \\ 0 \end{bmatrix}$　$X_1 = \begin{bmatrix} 0 \\ 1 \end{bmatrix}$

$\begin{bmatrix} 0 & 0 & 1 & 1 & 0 & 0 & & 0 & 0 & 0 & 0 & 0 & 0 & & 0 & 0 & 0 & 0 & 1 & 1 \\ 0 & 0 & 0 & 0 & 0 & 0 & & 0 & 0 & 0 & 0 & 0 & 0 & & 0 & 0 & 0 & 0 & 0 & 0 \\ 0 & 0 & 0 & 0 & 0 & 0 & & 0 & 0 & 0 & 0 & 0 & 0 & & 0 & 0 & 0 & 0 & 0 & 0 \\ 0 & 0 & 1 & 1 & 0 & 0 & & 0 & 0 & 0 & 0 & 0 & 0 & & 0 & 0 & 0 & 0 & 1 & 1 \end{bmatrix}$

$Y_2$=[ 0　0　0　0　1　1 ] , $Z_2$=[ 0　0　0　0　1　1 ]

$X_0 = \begin{bmatrix} 1 \\ 0 \end{bmatrix}$　$X_1 = \begin{bmatrix} 0 \\ 1 \end{bmatrix}$

$\begin{bmatrix} 0 & 0 & 0 & 0 & 1 & 1 & & 0 & 0 & 0 & 0 & 0 & 0 & & 0 & 0 & 0 & 0 & 1 & 1 \\ 0 & 0 & 0 & 0 & 0 & 0 & & 0 & 0 & 0 & 0 & 0 & 0 & & 0 & 0 & 0 & 0 & 0 & 0 \\ 0 & 0 & 0 & 0 & 0 & 0 & & 0 & 0 & 0 & 0 & 0 & 0 & & 0 & 0 & 0 & 0 & 0 & 0 \\ 0 & 0 & 0 & 0 & 1 & 1 & & 0 & 0 & 0 & 0 & 0 & 0 & & 0 & 0 & 0 & 0 & 1 & 1 \end{bmatrix}$

Hence, $A_{g,h,l}$ can be expressed as [10,11]:

$$A_{g,h,l} = \begin{bmatrix} b_{1,1,1} & b_{1,2,1} & \cdots & b_{1,N,1} & b_{1,1,2} & \cdots & \cdots & b_{1,N,2} & \cdots & b_{1,1,P} & b_{1,2,P} & \cdots & b_{1,N,P} \\ b_{2,1,1} & b_{2,2,1} & \cdots & b_{2,N,1} & b_{2,1,2} & \cdots & \cdots & b_{2,N,2} & \cdots & b_{2,1,P} & b_{2,2,P} & \cdots & b_{2,N,P} \\ \vdots & \vdots & \vdots & \vdots & \vdots & \vdots & \vdots & \vdots & \vdots & \vdots & \vdots & \vdots & \vdots \\ b_{M,1,1} & b_{M,2,1} & \cdots & b_{M,N,1} & b_{M,1,2} & \cdots & \cdots & b_{M,N,2} & \cdots & b_{M,1,P} & b_{M,2,P} & \cdots & b_{M,N,P} \end{bmatrix} \tag{15}$$

Consequently, the 3D-VWZCC cross-correlation property between $A^{(d)}$ and $A_{g,h,l}$ is expressed by the following equation, where $a^{(d)}{}_{i,j,l}$ is the component of $A^{(d)}$ and $b_{i,j,l}$ is the components of $A_{g,h,l}$ [10–12].

$$R^{(d)}(g,h,l) = \sum_{i=0}^{M-1} \sum_{j=0}^{N-1} \sum_{l=0}^{P-1} a^{(d)}{}_{i,j,l} b_{i,j,l} \tag{16}$$

Further, Table 4 shows the results of the cross-correlation of the 3D-VWZCC code and contains eight cases according to the characteristic matrices $A^{(d)}$. According to Table 3, all cases give the same cross-correlation value, taking into account the intersection of g, h and l. We notice that the values of cases 0 to 3 are the same values of cases 4 to 7, and this is due to the orthogonality of the VWZCC codes. Moreover, the value $W_1.W_2.W_3$ is varied and depends on the weight $W_1$, $W_2$ and $W_3$ of each spectral, temporal, and spatial component, respectively; as a result, the 3D-VWZCC code has good flexibility.

By compensating the characteristic matrices $A^{(d)}$ of Equation (1) in Equation (2), we obtain Table 4, which gives the values of $R^{(d)}$ that are equal to zero, except that $R^{(0)}$ and $R^{(4)}$ are written by the following equations:

$$R^{(0)}(g,h,l) = \begin{cases} 1 & if \ \ g = 0 \cap h = 0 \cap l = 0 \\ 0 & else \end{cases} \tag{17}$$

$$R^{(4)}(g,h,l) = \begin{cases} 1 & if \ \ g = 0 \cap h = 0 \cap l \neq 0 \\ 0 & else \end{cases} \tag{18}$$

**Table 4.** Cross correlation values of 3D-VWZCC code.

| | $R^{(0)}(g,h,l)$ | $R^{(1)}(g,h,l)$ | $R^{(2)}(g,h,l)$ | $R^{(3)}(g,h,l)$ |
|---|---|---|---|---|
| $g = 0 \cap h = 0 \cap l = 0$ | $W_1.W_2.W_3$ | 0 | 0 | 0 |
| $g = 0 \cap h \neq 0 \cap l = 0$ | 0 | $W_1.W_2.W_3$ | 0 | 0 |
| $g \neq 0 \cap h = 0 \cap l = 0$ | 0 | 0 | $W_1.W_2.W_3$ | 0 |
| $g \neq 0 \cap h \neq 0 \cap l = 0$ | 0 | 0 | 0 | $W_1.W_2.W_3$ |
| $g = 0 \cap h = 0 \cap l \neq 0$ | 0 | 0 | 0 | 0 |
| $g = 0 \cap h \neq 0 \cap l \neq 0$ | 0 | 0 | 0 | 0 |
| $g \neq 0 \cap h = 0 \cap l \neq 0$ | 0 | 0 | 0 | 0 |
| $g \neq 0 \cap h \neq 0 \cap l \neq 0$ | 0 | 0 | 0 | 0 |
| | $R^{(4)}(g,h,l)$ | $R^{(5)}(g,h,l)$ | $R^{(6)}(g,h,l)$ | $R^{(7)}(g,h,l)$ |
| $g = 0 \cap h = 0 \cap l = 0$ | 0 | 0 | 0 | 0 |
| $g = 0 \cap h \neq 0 \cap l = 0$ | 0 | 0 | 0 | 0 |
| $g \neq 0 \cap h = 0 \cap l = 0$ | 0 | 0 | 0 | 0 |
| $g \neq 0 \cap h \neq 0 \cap l = 0$ | 0 | 0 | 0 | 0 |
| $g = 0 \cap h = 0 \cap l \neq 0$ | $W_1.W_2.W_3$ | 0 | 0 | 0 |
| $g = 0 \cap h \neq 0 \cap l \neq 0$ | 0 | $W_1.W_2.W_3$ | 0 | 0 |
| $g \neq 0 \cap h = 0 \cap l \neq 0$ | 0 | 0 | $W_1.W_2.W_3$ | 0 |
| $g \neq 0 \cap h \neq 0 \cap l \neq 0$ | 0 | 0 | 0 | $W_1.W_2.W_3$ |

## 4. 3D-VWZCC System Description

The incoherent OCDMA transceiver system based on the suggested three-dimensional VWZCC code is presented in Figure 2; this system is composed of four essential blocks: data generation unit and optical source, 3D-VWZCC encoder, 3D-VWZCC decoder, and information retrieval unit. Moreover, the system contains K ($K_1 K_2 K_3$) combinations of transceivers and P star couplers, while each subscriber is assigned by the signature 3D-VWZCC code word $A_{g,h,l}$; the system is described as follows: the transmitter part (Figure 2a) contains an information generator (image, video, sound, etc.), on/off keying modulator, electrical/optical modulator, incoherent optical source, M spectral components of fiber Bragg grating FBGs ($FBG_1$ and $FBG_2$), N temporal components, and P spatial components.

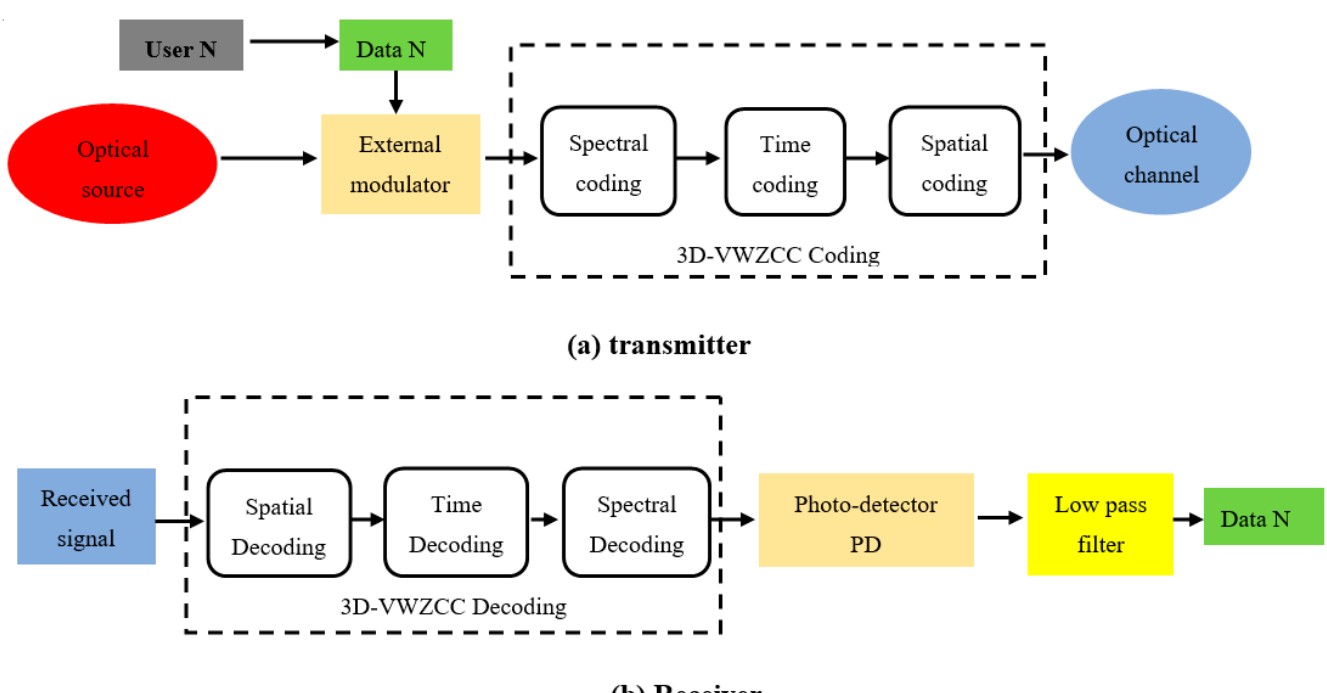

**Figure 2.** 3D-VWZCC OCDMA transceiver system. (**a**) Transmitter; (**b**) Receiver.

In the first phase, the data coming out of the information source are modulated by on/off keying, then combined with an incoherent optical source and converted into the optical domain using an external MZM modulator (Mach-Zehnder modulator); then, the optical pulses are transmitted to the fiber Bragg gratings ($FBG_1$ et $FBG_2$). The first time, the optical pulses are encoded by the $FBG_1$ according to the spectral sequence $X_g$, where the encoder $FBG_1$ encoder selects specific wavelengths and reflects them, whereas the others that are different from (1) are eliminated. The second time, the selected wavelengths are sent to the $FBG_2$ spectral encoder to compensate for the run-trip delay. We note that $FBG_2$ has the same number of gratings as $FBG_1$ but with reversed arrangements.

In the second phase, the wavelengths leaving the spectral encoder $FBG_2$ are sent to the delay lines to apply the time spread according to the time sequence $Y_h$, where the input of the delay lines is divided into $W_2$ equal parts. Furthermore, once the time coding is complete, the optical signals are multiplexed and propagated to the demultiplexer for spatial coding.

In the third phase, the pulses in the output of the demultiplexer are divided into identical versions equal to $W_3$ and are connected to the star couplers to perform the spatial coding according to the sequence $Z_l$. In this case, the optical signal is coded in the 3D-VWZCC code.

The design of the 3D-VWZCC receiver is shown in Figure 2b, and contains several components: optical signal multiplexer (combiner), correlator, two sets of fiber Bragg gratings FBGs ($FBG_1$ and $FBG_2$), photodiodes PDs, and low pass filters (LPFs). The process of receiver that is the reverse of transmitter is described as follows: the optical signals received by the couplers are combined by the multiplexer according to $Z_l$. and shared to $W_2$ equal signals by the correlator and connected with the delay lines for the time decoding according to $Y_h$, then the pulses at the output of the time delay lines are connected with fiber Bragg gratings FBGs for spectral decoding, where in the first step, the pulses are passed to $FBG_1$ to select specific wavelengths according to the sequence $X_g$, and filtered the others. Then, the selected pulses are reflected and sent to $FBG_2$ to compensate for the delays introduced due to the $FBG_1$ encoding. As a result, three-dimensional decoding has been performed. After that, the resulting optical signal is passed through to the photo-detector PD to convert the optical signal to an electrical signal in a direct way due to the ZCC

property, and finally, the low pass Bessel filter (LPF) is used to extract the desired data according to the data sent by the transmitter.

## 5. 3D-VWZCC System Performance

In order to analyze and evaluate the performance of our system, two essential parameters are used: bit error rate (BER), which is defined by the ratio between the number of erroneous bits and the total number of transmitted bits, and signal-to-noise ratio (SNR), which is also defined by the ratio between the total received power and the noise power that affected the system. In addition, the system is affected by three types of noise: thermal noise, shot noise, and PIIN noise; further, our code has the property of zero cross-correlation ZCC. Therefore, we take into account the impact of these noises to calculate the signal-to-noise ratio, with the total photocurrent noise expressed as [10,11,13,15]:

$$\left\langle \sigma_{noise}^2 \right\rangle = \left\langle \sigma_{Thermal}^2 \right\rangle + \left\langle \sigma_{Shot}^2 \right\rangle + \left\langle \sigma_{PIIN}^2 \right\rangle \tag{19}$$

$$\sigma_{noise}^2 \frac{4K_b T_n B_r}{R_l} + 2eB_r I + B_r I^2 \tau_c \tag{20}$$

where $K_b$ is Boltzmann's constant, $T_n$ is the absolute temperature in Kelvin, $B_r$ is the electrical bandwidth, $R_l$, $e$, $I$ are load resistance, electron charge, and variance of the receiver photocurrent, respectively, and $\tau_c$ denotes the light coherence time, which is defined by the following equation where $G$ is the single sideband power spectral density (PSD) [10–13]:

$$\tau_C = \frac{\int_0^{+\infty} G^2(v)dv}{\left[\int_0^{+\infty} G(v)dv\right]^2} \tag{21}$$

To simplify the analysis of 3D-VWZCC, we take into consideration four hypotheses presented as follows: first, the incoherent broadband light source has a non-variable spectrum during the light bandwidth $[v_0 - \Delta v/2 , v_0 + \Delta v/2]$, where $v_0$ is the central frequency and $\Delta v$ is the bandwidth of the light source. Second, the spectral width is equi-spectro for all users. Third, equal power for all users at the receiver level. Finally, the fourth is each bit data stream of each user is synchronized.

According to the above assumptions, the power spectral density (PSD) of the received signals $r(v)$ can be written as [10–13]:

$$r(v) = \frac{P_{sr}}{W_2 W_3 \Delta v} \sum_{k=1}^{K} d_k \sum_{i=0}^{M-1} \sum_{j=0}^{N-1} \sum_{l=0}^{P-1} a_{i,j,l} \prod(v,i) \tag{22}$$

$P_{sr}, \Delta v, W_2, W_3$ and $\prod(v,i)$ are the effective received power, bandwidth of the light source, weight of temporal sequence, and weight of spatial sequence, respectively, and $d_k$ is the data bit of the $k^{th}$ user, which can be "1" or "0"and $\prod(v,i)$ defined as:

$$\prod(v,i) = \left\{ u\left[v - v_0 - \frac{\Delta v}{2M}(-2M + 2i)\right] - \left[v - v_0 - \frac{\Delta v}{2M}(-M + 2i + 2)\right] \right\} = u\left[\frac{\Delta v}{M}\right] \tag{23}$$

$u[v]$ is a unit step function given as:

$$u(v) = \begin{cases} 1 & when \; v \in [0, +\infty[ \\ 0 & when \; v \in [-\infty, 0] \end{cases} \tag{24}$$

The output of the currents from the photodiode (PD) to the receiver is according to the cross-correlation between $A^{(d)}_{0,0,0,}$ and $A_{0,0,0}$ of the code 3D-VWZCC denoted by $I_r$ and can be written as:

$$I = \mathcal{R} \int_0^{+\infty} r(v)dv = \mathcal{R} \int_0^{+\infty} \frac{P_{sr}}{W_2 W_3 \Delta v} \sum_{k=1}^{K} d_k R_{i,j,l}^{(d)} \prod(v,i)dv \tag{25}$$

By substituting Equations (17) and (23) into Equation (25) we find:

$$I = \frac{\mathcal{R}P_{sr}}{W_2 W_3 \Delta v}\left[W_1 W_2 W_3 + \sum_{k=1}^{K} d_k R_{0,0,0}^{(0)}\right] \times \frac{\Delta v}{M} = \frac{\mathcal{R}P_{sr}W_1}{M} \tag{26}$$

where $\mathcal{R}$ is the photodiode responsivity given by $\mathcal{R} = ne/hv_0$, $\eta$ is the quantum efficiency of the photo-diode, and $h$ is Plank's constant and $e$ is the electron charge. $M$, $W_1$ are code length and code weight for spectral sequence $X_g$, respectively. For $M = W_1 K_1$ and $K = K_1 K_2 K_3$, the output of the currents from the photodiode (PD) depends on $K$ and $K_2 K_3$, and can be written as:

$$I = \frac{\mathcal{R}P_{sr}K_2 K_3}{K} \tag{27}$$

The total variance of phase-induced intensity noise PIIN current, expressed as follows:

$$\sigma_{PIIN}^2 = B_r \mathcal{R}^2 \int_0^{+\infty} G^2(v)dv \tag{28}$$

$$\sigma_{PIIN}^2 = B_r \mathcal{R}^2 \left[\int_0^{+\infty} \frac{P_{sr}}{W_2 W_3 \Delta v} \sum_{k=1}^{K} d_k \sum_{i=0}^{M-1} \sum_{j=0}^{N-1} \sum_{l=0}^{P-1} a_{i,j,l} \prod(v,i)dv\right]^2$$

$$\sigma_{PIIN}^2 = \frac{B_r \mathcal{R}^2 P_{sr}{}^2}{(W_2 W_3 \Delta v)^2}\left[(W_1 W_2 W_3)^2 \times \frac{\Delta v}{M}\right] = \frac{B_r \mathcal{R}^2 P_{sr}{}^2 W_1{}^2}{\Delta v M} = \frac{M B_r}{\Delta v}\left[\frac{\mathcal{R}P_{sr}W_1}{M}\right]^2 = \frac{M B_r I^2}{\Delta v} \tag{29}$$

Shot noise is given by the following relation according to the output currents from the photodiode (PD):

$$\sigma_{shot}^2 = 2eB_r I \tag{30}$$

$$\sigma_{shot}^2 = 2eB_r\left[\frac{\mathcal{R}P_{sr}W_1}{M}\right] = 2eB_r\left[\frac{\mathcal{R}P_{sr}K_2 K_3}{K}\right] \tag{31}$$

By substituting Equations (29) and (31) into Equation (20), we obtain:

$$\sigma_{noise}^2 = \frac{4K_b T_n B_r}{R_l} + 2eB_r I + B_r I^2 \tau_c \tag{32}$$

$$\sigma_{noise}^2 = \frac{4K_b T_n B_r}{R_l} + 2eB_r\left[\frac{\mathcal{R}P_{sr}W_1}{M}\right] + \frac{B_r \mathcal{R}^2 P_{sr}{}^2 W_1{}^2}{\Delta v M}$$

$$\sigma_{noise}^2 = \frac{4K_b T_n B_r}{R_l} + 2eB_r\left[\frac{\mathcal{R}P_{sr}K_2 K_3}{K}\right] + \frac{B_r \mathcal{R}^2 P_{sr}{}^2 W_1 K_2 K_3}{\Delta v K} \tag{33}$$

The probability of sending "1" and "0" for each user is equi-probable, hence, Equation (33) becomes:

$$\sigma_{noise}^2 = \frac{4K_b T_n B_r}{R_l} + eB_r\left[\frac{\mathcal{R}P_{sr}K_2 K_3}{K}\right] + \frac{B_r \mathcal{R}^2 P_{sr}{}^2 W_1 K_2 K_3}{2\Delta v K} \tag{34}$$

Depending on Equations (27) and (34), the average *SNR* at receiver can be written as:

$$SNR = \frac{I^2}{\langle\sigma_{noise}^2\rangle} = \frac{\left(\frac{\mathcal{R}P_{sr}K_2 K_3}{K}\right)^2}{\frac{4K_b T_n B_r}{R_L} + eB_r\left[\frac{\mathcal{R}P_{sr}K_2 K_3}{K}\right] + \frac{B_r \mathcal{R}^2 P_{sr}{}^2 W_1 K_2 K_3}{2\Delta v K}} \tag{35}$$

Finally, by using Gaussian approximation, the bit error rate is derived from *SNR* and given by the following Equation [10–13,29]:

$$BER = \frac{1}{2} \times erfc\sqrt{SNR/8} \tag{36}$$

By substituting Equation (35), into Equation (36) we obtain:

$$BER = \frac{1}{2} \times erfc \sqrt{\frac{\left(\frac{\mathcal{R} P_{sr} K_2 K_3}{K}\right)^2}{8 \times \left[\frac{4 K_b T_n B_r}{R_L} + e B_r \left[\frac{\mathcal{R} P_{sr} K_2 K_3}{K}\right] + \frac{B_r \mathcal{R}^2 P_{sr}^2 W_1 K_2 K_3}{2 \Delta v K}\right]}} \tag{37}$$

Where *erfc(a)* is defined as:

$$erfc(a) = \frac{2}{\sqrt{\pi}} \int_X^{+\infty} e^{-(X^2)} dX \tag{38}$$

## 6. Numerical Results and Discussion

In this phase, the performance of the 3D-VWZCC code is compared to the 3D-perfect difference (3D-PD), 3D-perfect difference/multi-diagonal (3D-PD/MD), and 3D-dynamic cyclic shift/multi-diagonal (3D-DCS/MD), which are previously published in [13–15] in terms of BER, SNR, and Q factor as a function of spectral width, number of active users, data rate, and effective source power. In addition, thermal noise, shot noise, and phase-induced intensity noise are taken into account during the simulation; in addition, Table 5 shows the parameters that are used in the MATLAB simulation with lengths $M = 7$, $N = 13$, and $P = 3$ for the spectral, temporal, and spatial components, respectively.

**Table 5.** The values of the parameters in the 3D-VWZCC system for numerical calculation.

| Parameters | Symbol | Value |
|---|---|---|
| Photo detector responsivity | $\mathcal{R}$ | 0.75 |
| Data rate | $R_b$ | 1.25 Gbp |
| Number of users | K | 150 |
| Electric bandwidth 1D, 2D | $B_{r1,2}$ | $0.5 \times R_b$ GHz |
| Electric bandwidth 3D | $B_{r3}$ | $0.5 \times N \times R_b$ GHz |
| Receiver load resistor | $R_l$ | 1030 Ω |
| Spectral width of light | $\Delta v$ | 5 THz |
| Effective source power | $P_{sr}$ | −10 dBm |
| Receiver noise temperature | $T_n$ | 300 K |
| Electron charge | E | $1.6 \times 10^{-19}$ c |
| Boltzman's constant | $K_b$ | $1.38 \times 10^{-23}$ J/K |

Figure 3 shows the variation of BER as a function of the number of simultaneous users, where the effective source power and data rate are equal to −10 dBm and 1.25 Gbps, respectively. When the property of the optical requirement is at $10^{-9}$, the code 3D-PD, 3D-PD/MD, and 3D-DCS/MD provides a user capacity of approximately 86, 98, and 109, respectively, while 1D-VWZCC, 2D-VWZCC, and 3D-VWZCC codes provide a significant number of users equal to 62, 141, and 317, respectively. Furthermore, our 3D-VWZCC code offers a system capacity of 3.686 times the 3D-PD code, 3.234 times the 3D-PD/MD code, and 2.908 times the 3D-DCS/MD code, while 3D-VWZCC offers a capacity of 5.112 times the 1D-VWZCC and 2.248 times the 2D-VWZCC for the same family of this code. Therefore, the results of our 3D-VWZCC system code demonstrate a superiority of 36.86%, 32.34%, and 29.09% compared to the codes 3D-PD, 3D-PD/MD, and 3D-DCS/MD, respectively. In addition, our code has the superiority of 51.12% and 22.48% of one-dimensional and two-dimensional VWZCC code for the same family of this code, respectively. As a result, we explain these results by a small impact of PIIN noise on our code compared to the codes in the three-dimensional domain (3D-PD, 3D-PD/MD, and 3D-DCS/MD) due to zero cross correlation and short code length compared to the 1D-VWZCC code, and less spatial components compared to 2D-VWZCC.

Figure 4 shows the variation of the SNR against number of simultaneous users for a data rate of 1.25 Gbps and a source power $P_{sr}$ of −10 dBm for each transmitter. We note

that all codes are affected by PIIN noise with different degrees depending on the code, except for the 1D-VWZCC code; the PIIN noise of this code is equal to zero due to zero cross correlation properties, for the same length (M = 7, N = 13, P = 3) of all codes except 1D-VWZCC and 2D-VWZCC. We set the number of users at 400, our suggested code has a very high signal to noise ratio, reaching 158, while 3D-PD, 3D-PD-MD, and 3D-DCS/MD codes have a signal to noise ratio up to 35.1, 35.3, and 51, respectively. On the other hand, the codes of the same family, 1D-VWZCC and 2D-VWZCC, offer an SNR up to 6 and 41, respectively. As a result, the 3D-VWZCC code provides an SNR up to 4.5 times the 3D-PD, 4.45 times the 3D-PD/MD, and 3.08 times the 3D-DCS/MD, whereas the 3D-VWZCC code provides an SNR up to 26.3 times the 1D-VWZCC and 3.85 times the 2D-VWZCC. Therefore, our code has a very high received power compared to the received noise power, and this is due to high immunity against PIIN noise thanks to the zero cross correlation property of this code compared to codes 3D-PD, 3D-PD/MD, and 3D-DCS/MD.

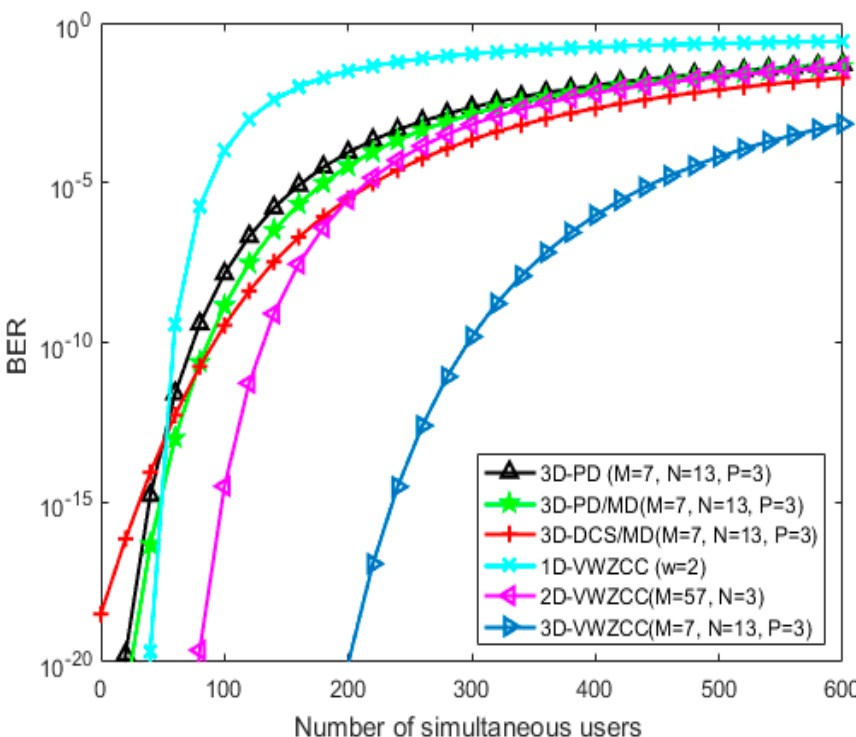

**Figure 3.** BER against number of simultaneous users when the effective power $P_{sr}$ = −10 dBm and data rate $R_b$ = 1.25 Gbps.

The variation of BER as a function spectral width is indicated in Figure 5 when the number of users equals 150 users and effective source power $P_{sr}$ equals to −10dBm. At the minimum optical transmission value $10^{-9}$, our proposed system needs a spectral width up to 0.15 THz, while the 3D-PD, 3D-PD/MD, 3D-DCS/MD, and 2D-VWZCC codes need a large spectral width up to 8 THz, 6.3 THz, 5.27 THz, and 3.75 THz, respectively. Therefore, the 3D-VWZCC code saved a spectral width around 8.85 THz from the 3D-PD code, 6.15 THz, 5.12 THz, and 3.6 THz from 3D-PD/MD code, 3D-DCS/MD, and 2D-VWZCC, respectively. Hence, we interpret these results by the zero cross correlation property of our code.

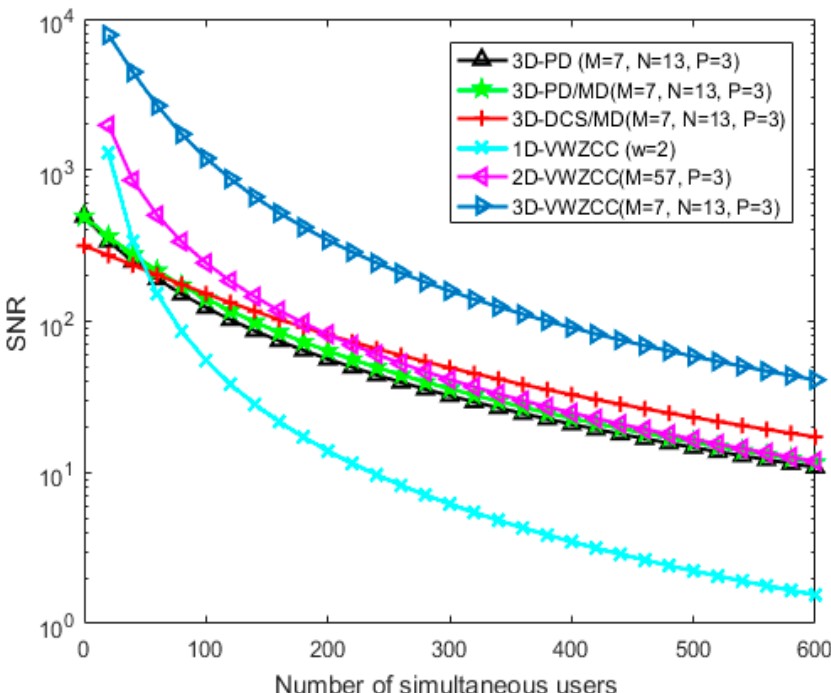

**Figure 4.** SNR against number of simultaneous users when the effective source power $P_{sr}$ = −10 dBm and data rate $R_b$ = 1.25 Gbps.

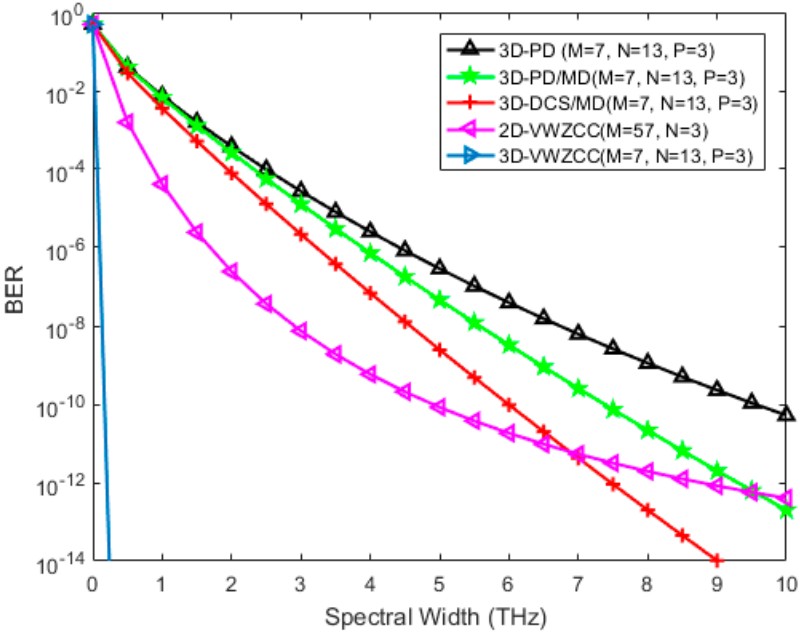

**Figure 5.** BER versus spectral width when the number of active users K = 150 and data rate $R_b$ = 1 Gbps.

Figure 6 shows the variation of BER against effective source power $P_{sr}$, where a number of simultaneous users equal to 150 and data rate of 1.25 Gbps, for $P_{sr}$ less than −30 dBm (<−30 dBm); we note that all the codes have the same BER, and at $P_{sr}$ greater than −30 dBm (>−30 dBm) we notice a significant variation in the value of BER. Consequently, in the acceptable BER $10^{-9}$ our code needs a power up to −11 dBm; on the other hand, the 2D-VWZCC and 1D-VWZCC code of the same code family accommodate an effective power of approximately −5.1 dBm and −3.96 dBm, respectively. In addition, the codes 3D-PD,

3D-PD/MD, and 3D-DCS/MD have power limited in BER greater than $10^{-5}$. Therefore, our 3D-VWZCC system gives a better performance and it can save the received power to about $-5.9$ dBm and 7.04 dBm compared to 2D-VWZCC and 1D-VWZCC; this is due to a low power consumption due to short length compared to a one dimensional (1D) code, two dimensional (2D) code, high immunity against PIIN noise, and multi-access interference MAI compared to 3D-PD, 3D-PD/MD, and 3D-DCS/MD.

Figure 7 shows the variation of BER versus data rate, where the number of simultaneous users equals 250 and spectral width equals 5THz. According to the figure, all codes provide a data rate less than 0.5 Gbps at an acceptable BER $10^{-9}$, and approximately 0.38 Gbps, 0.4 Gbps, 0.51 Gbps, and 0.55 Gbps for 3D-PD, 3D-PD/MD, 2D-VWZCC, and 3D-DCS/MD codes, respectively, except for the 1D-VWZCC code, which can provide data rate up to 0.2 Gbps at a BER equal to $10^{-4}$. Further, when increasing the data rate, the 3D-VWZCC code is the only code that meets the optical transmission requirements with a data rate reaching 2 Gbps. In addition, the 3D-VWZCC system improved the data capacity up to 5.26, 5, and 3.62 times of 3D-PD, 3D-PD/MD, and 3D-DCS/MD code, respectively, whereas the 3D-VWZCC code increased the transmission capacity up to 3.62 times of the passage from two dimension (2D) to the three dimension (3D). As a result, we demonstrate these results by our code's high capacity and immunity against MAI and the effect of PIIN noise.

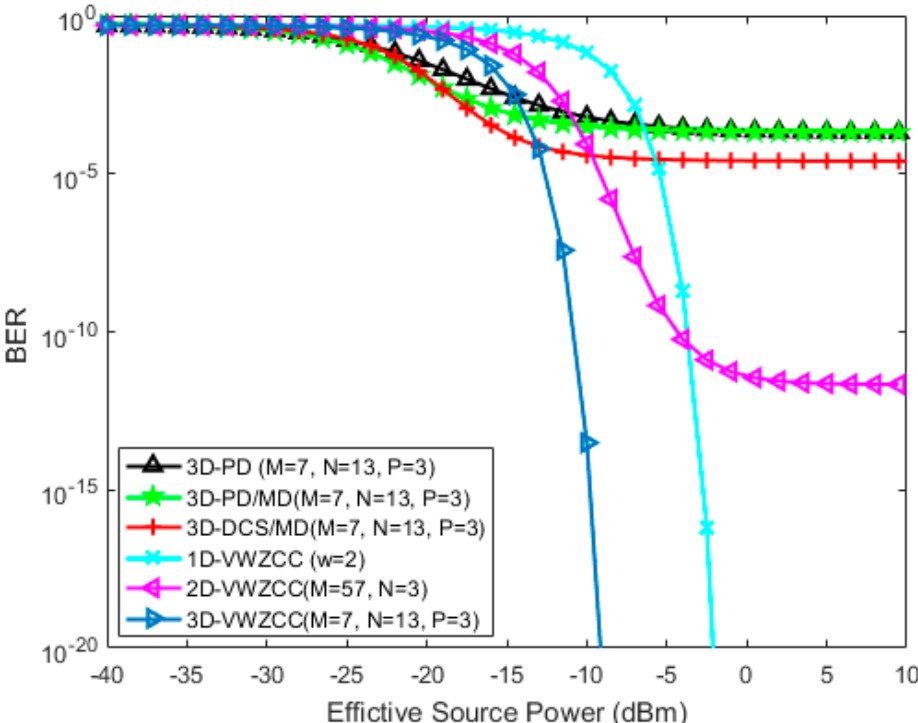

**Figure 6.** BER against effective source power when the number of simultaneous users K = 150 and data rate $R_b$ = 1.25 Gbps.

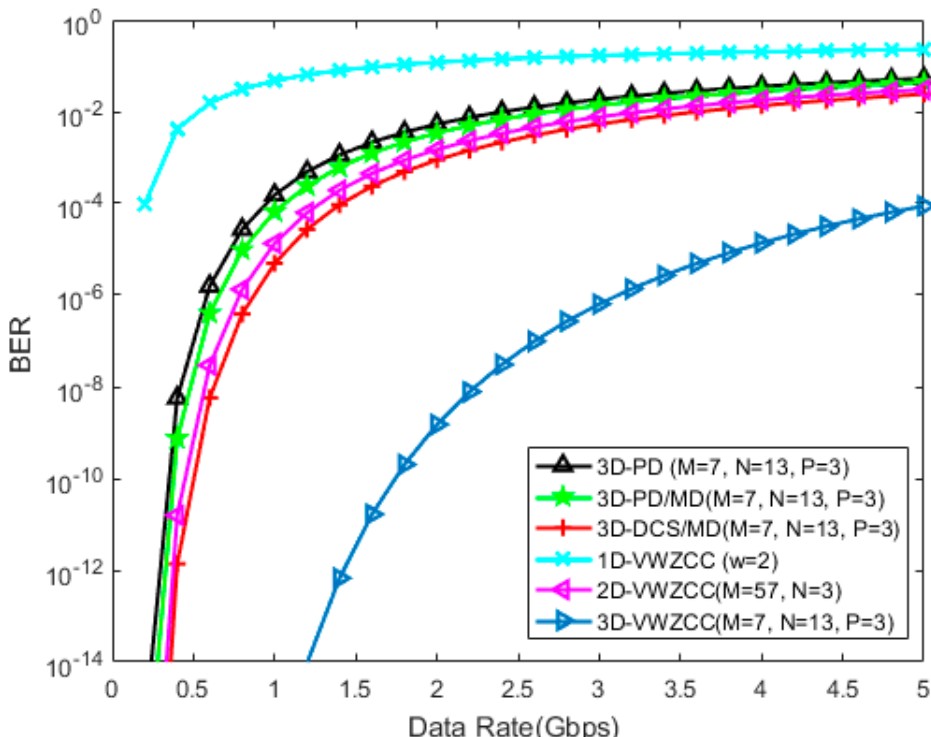

**Figure 7.** BER versus data rate when the number of active users K = 250 and effective source power P$_{sr}$ = −10 dBm.

Figure 8 shows the variation of BER as a function effective source power where our 3D-VWZCC code is tested with various noises, with the data rate and number of simultaneous users equal to 1.25 Gbps and 250, respectively. According to the figure, the code is tested with three types of noise; the blue curve presents a combination between thermal noise, shot noise, and PIIN noise, the green curve presents a combination between shot noise and PIIN noise, and the red curve presents a combination between thermal noise and PIIN noise. It is clear that the impact of combination between thermal noise, shot noise, and PIIN noise on the system have the same effect as the combination between thermal noise and PIIN noise. In contrast, the combination between shot noise and PIIN noise demonstrates a significant variation compared to the other curves. As a result, the impact of shot noise is totally neglected and the system is mainly affected by thermal noise and PIIN noise.

The variation of photocurrent noise against effective source power, where the data rate and number of simultaneous users are equal to 1.25 Gbps and 250 users, respectively, is shown in Figure 9 when the effective source power is lower than −10 dBm. All codes have a stable value reaching $10^{-13}$, and during the increase in the effective source power we notice an important variation in the photocurrent noise value between 3D-PD, 3D-PD/MD, and 3D-DCS/MD, as well as 3D-MD and 3D-VWZCC; this is due to the characteristic of zero cross correlation, which reduces the effect of MAI and PIIN noise and direct detection at the receiver. Consequently, the shot noise becomes inefficient in the system and PIIN noise becomes the main noise, which degrades the performance of the optical multiplexing system.

Figure 10 shows the variation of phase-induced intensity noise PIIN as a function of received power, where all the codes have the same length (M = 7, N = 13, P = 3) and data rate, and the number of simultaneous users are equal to 1.25Gbps and 250 users, respectively. It is clear that our 3D-VWZCC code has less PIIN noise power than 3D-PD, 3D-PD/MD, and 3D-DCS/MD codes. Thus, we interpret these results by the zero cross correlation property of the VWZCC code, which requires a simple detection method that reduces the effect of PIIN compared to other codes.

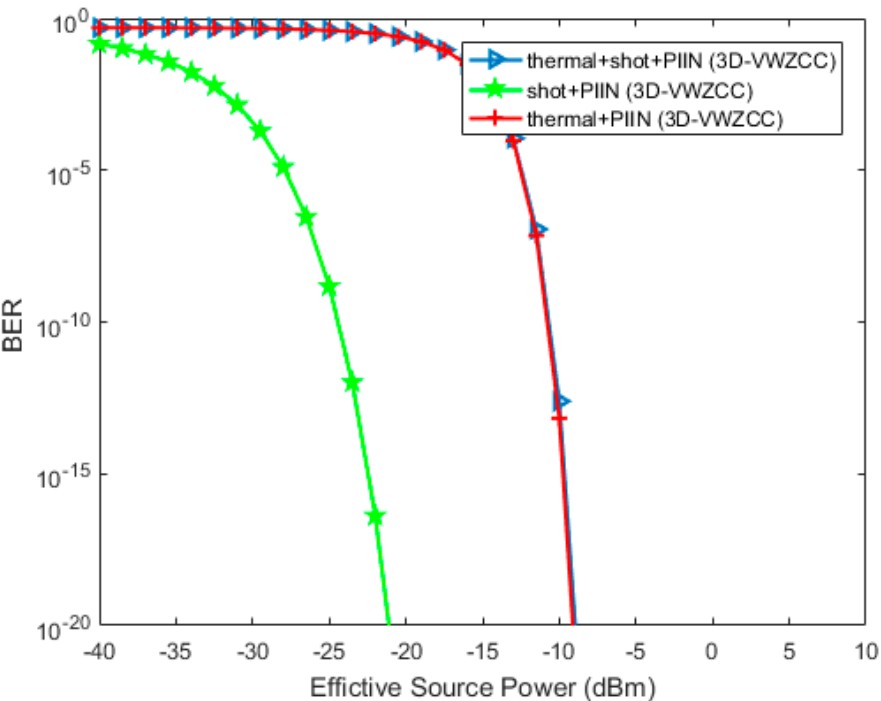

**Figure 8.** BER against effective source power when the number of simultaneous users K = 250 and data rate $R_b$ = 1.25 Gbps.

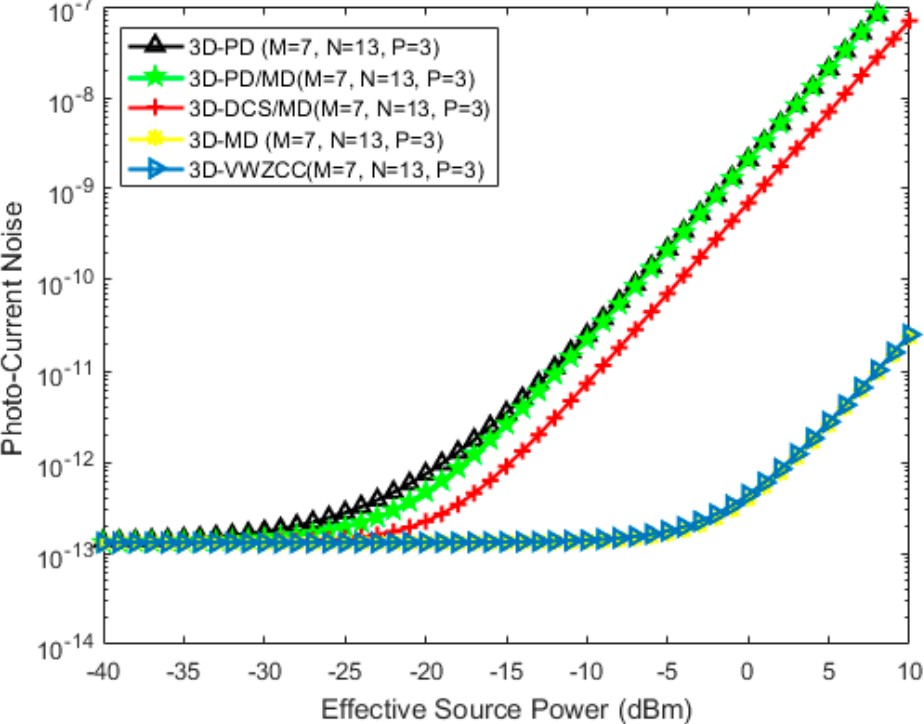

**Figure 9.** Photo current noise against effective source power when the number of simultaneous users K = 250 and data rate $R_b$ = 1.25 Gbps.

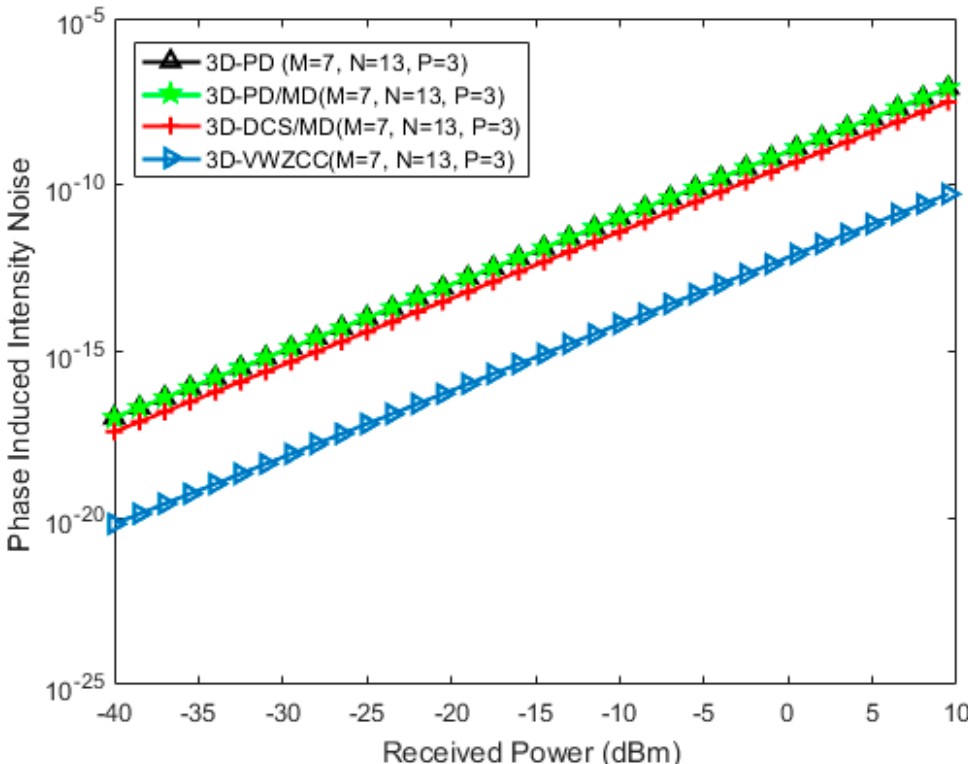

**Figure 10.** Phase-induced intensity noise against received power when the number of simultaneous users K = 250 and data rate R$_b$ = 1.25 Gbps.

Figure 11 indicates the variation of SNR of our code against the number of simultaneous users with the impact of different combinations of noises, thermal noise + shot noise + PIIN noise, shot noise + PIIN noise, and thermal noise + PIIN noise, where the effective source power equal to $P_{sr}$ = −10 dBm and the data rate $R_b$ = 1.25 Gbps. We observe that our code is mainly affected by thermal noise and PIIN noise, to a lesser degree, and shot noise is neglected due to the thermal agitation of electrons in electronic components; this is what has been proven in Figures 8–10.

Figure 12 indicates the variation of the Q factor against the number of simultaneous users where the effective source power equals −10 dBm and the data rate equals 1.25 Gbps. At the value of the quality factor Q satisfying the optical transmission requirements (Q = 6), all codes do not exceed number of users greater than 200 except our code 3D-VWZCC; as a result, the number of users can reach 83, 96, 106 for the codes 3D-PD, 3D-PD/MD, 3D-DCS/MD, respectively, and the passage from 1D and 2D to 3D for our code accommodates a number of users equal to 61,140, and 312, respectively. Hence, the 3D-VWZCC code increased the cardinality by 3.75, 3.25, 2.94, 5.11, and 2.15 times compared to the 3D-PD, 3D-PD/MD, 3D-DCS/MD, 1D-VWZCC, and 2D-VWZCC codes, respectively. Therefore, we explain the superiority of our code compared to others by high multiplexing capacity due to the immunity against the MAI, which allows supporting a very high number of users at higher data rate.

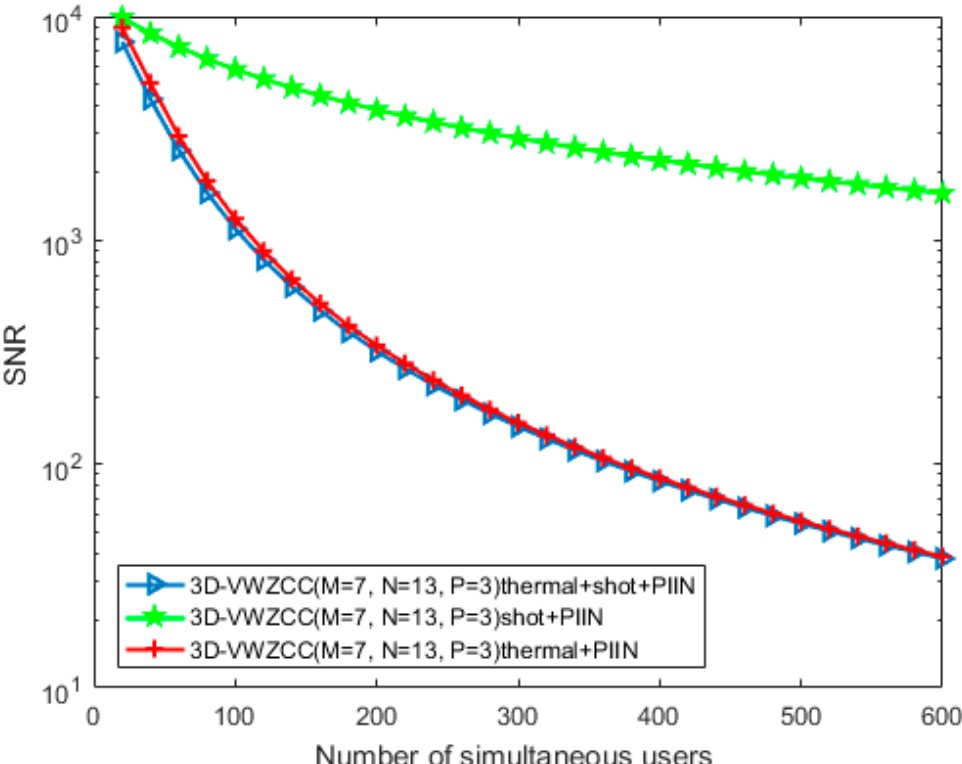

**Figure 11.** SNR versus number of simultaneous users when effective source power $P_{sr}$ = −10 dBm and data rate $R_b$ = 1.25 Gbps.

Furthermore, the parameters of BER and SNR are considered as the most conclusive factors of merit for evaluating the performance of the optical system; however, another parameter is taken into consideration in order to reinforce the results obtained. The Error Vector Magnitude (EVM) is taken into account in recent years in order to measure the quality of the signals transmitted in optical communication systems; it compares the received data with the original data. Furthermore, the lower the EVM, the better the signal quality. In our work, the *EVM* is estimated as a function of *SNR*, as indicated in the equation below [30,31].

$$EVM \ (\%) = \left( \sqrt{\frac{1}{SNR}} \right) \times 100 \tag{39}$$

As shown in Figure 13, the variation of *EVM* as a function of the number of users was studied, where the results are obtained at a data rate of 1.25 Gbps and an effective source power of −10 dBm. It is clear that our 3D-VWZCC code has a very low EVM value compared to the other codes. At acceptable EVM (EVM = 10%), our code supports a large number of users more than 300 users, whereas 3D-PD, 3D-PD/MD, 3D-DCS/MD, 2D-VWZCC, and 1D-VWZCC codes support 125, 142, 180, 185, and 74 users. As a result, we explain this superiority by the ZCC property of our code, which perfectly eliminates the multiple access interference (MAI) and reduces the effect of PIIN noise.

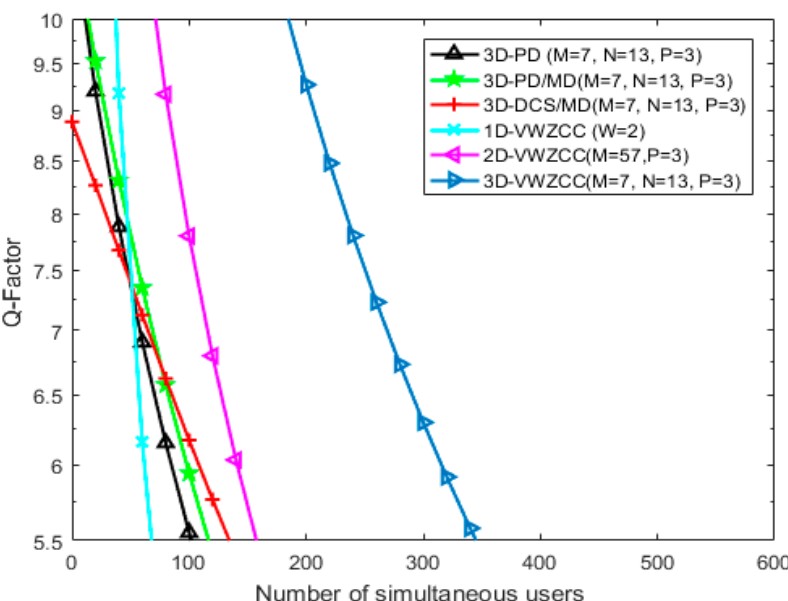

**Figure 12.** Q factor versus number of simultaneous users when effective source power $P_{sr} = -10$ dBm and data rate $R_b$ = 1.25 Gbps.

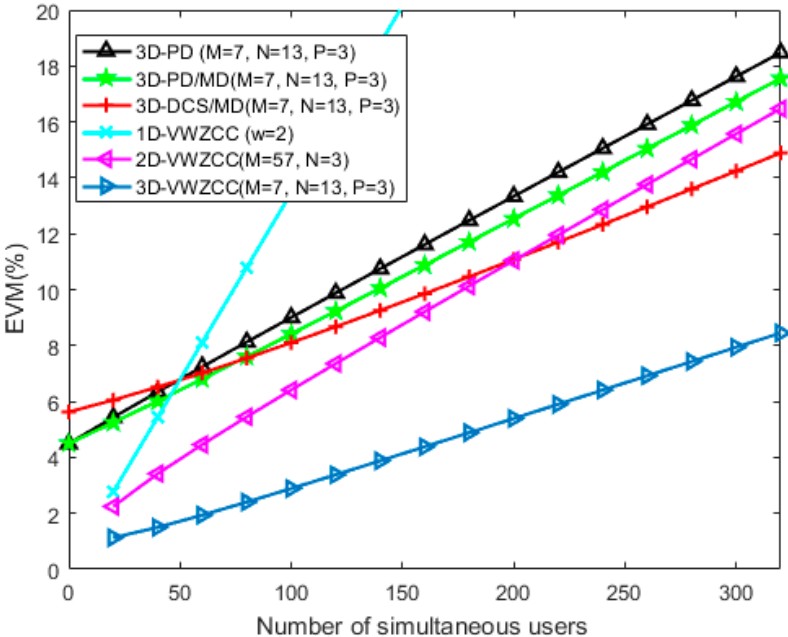

**Figure 13.** EVM versus number of simultaneous users when effective source power $P_{sr} = -10$ dBm and data rate $R_b$ = 1.25 Gbps.

Figure 14 shows the variation of EVM as a function of the data rate where the number of users and the effective source power are fixed at 250 and −10 dBm, respectively. It is clear that the proposed code has a low percentage of EVM despite the high data rate compared to the 3D-PD, 3D-PD/MD, 3D-DCS/MD, 2D-VWZCC, and 1D-VWZCC codes. It accommodates a data rate above 2.5 Gbps with its allowable EVM value (10%), while 3D-PD, 3D-PD/MD, 3D-DCS/MD, 2D-VWZCC, and 1D-VWZCC codes are accommodating 0.52 Gbps, 0.59 Gbps, 0.75 Gbps, 0.71 Gbps, and 0.15 Gbps, respectively. We explain these results by the fact that our code shows a low error rate at the receiver level, which results in a low EVM value due to low PIIN noise and short code length compared to other codes.

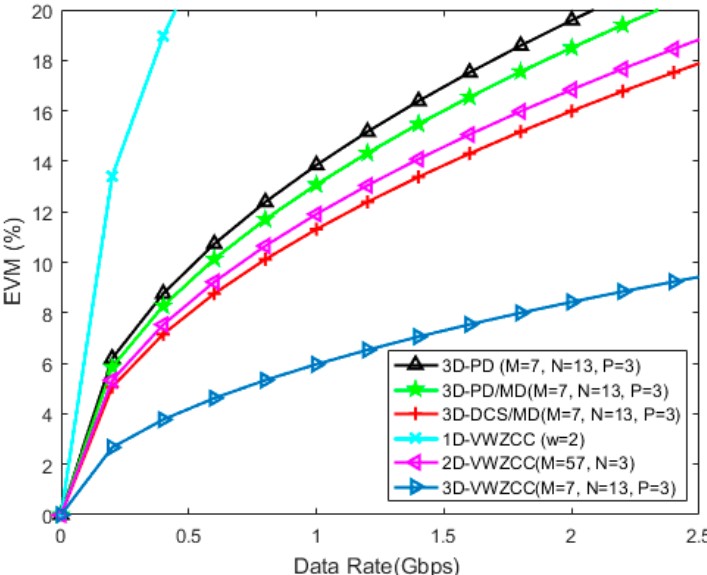

**Figure 14.** EVM versus data rate when the number of active users K = 250 and effective source power $P_{sr}$ = −10 dBm.

## 7. Network Simulation and Discussion

In the second phase, in order to prove the results obtained in the first phase of the simulation by MATLAB software, our system, simulated using Optisystem software, is shown in Figure 15 for eight users; further, Table 6 shows the parameters that are used during the simulation. Moreover, Table 7 presents the coding process used in the simulation using Optisystem software, where the table is divided into three parts: spectral coding Y, which is performed by fiber Bragg gratings (FBG); temporal coding Y, which is performed by the delay lines; and spatial coding Z, which is performed by the star couplers. Moreover, the process is carried out according to the proposed code.

**Table 6.** Simulation parameters of a 3D-VWZCC network.

| Parameters | Value |
|---|---|
| Data rate | 1 and 2 Gbps |
| Effective source power | −115 dBm |
| Central wavelength | 1552.5 nm |
| FBG bandwidth | 0.8 nm |
| MZM extinction ratio | 30 dB |
| PD responsivity | 1 A/w |
| Dark current | 5 nA |
| Thermal noise | $1.8 \times 10^{-23}$ W/Hz |
| Coupling coefficient | 0.5 |
| Cutoff frequency | $R_b \times 0.75$ GHz |
| Number of users | 08 |
| Code weight | 2 |
| Time delay | 0 ns, 0.25 ns, 0.5 ns, and 0.75 ns |
| Shot noise distribution | Gaussian |

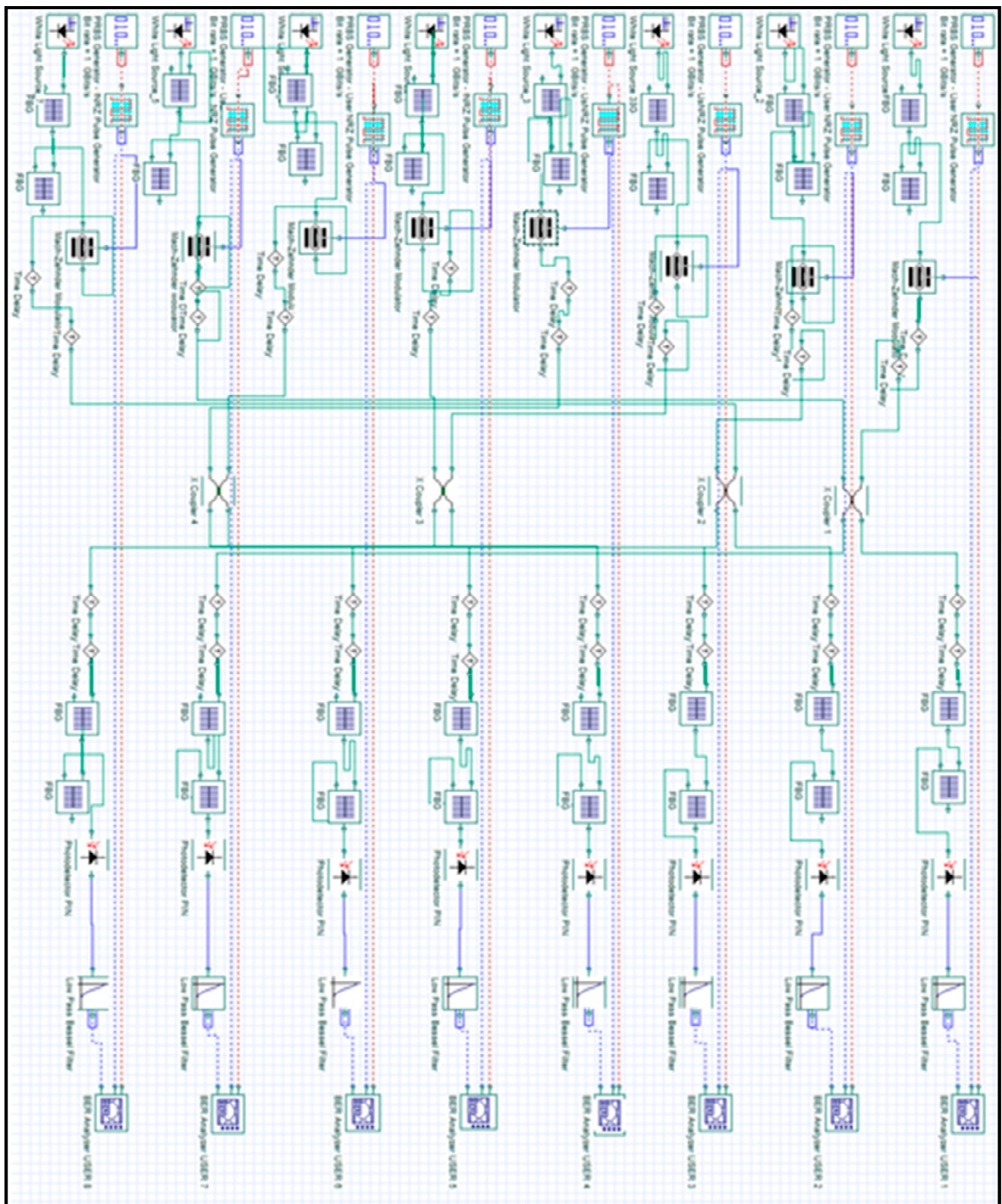

**Figure 15.** Diagram block of the 3D-VWZCC/OCDMA system.

**Table 7.** Implemented code words for network simulation.

| Subcarriers | $X_g^T$ | | | | $Y_h$ | | | | $Z_L$ | | | | 3D-VWZCC |
|---|---|---|---|---|---|---|---|---|---|---|---|---|---|
| | $\lambda_1$ | $\lambda_1$ | $\lambda_3$ | $\lambda_4$ | $t_1$ | $t_2$ | $t_3$ | $t_4$ | $C_1$ | $C_2$ | $C_3$ | $C_4$ | |
| $U_1$ | | | | | | | | | | | | | |
| $U_2$ | | | | | | | | | | | | | |
| $U_3$ | | | | | | | | | | | | | |
| $U_4$ | | | | | | | | | | | | | |
| $U_5$ | | | | | | | | | | | | | |
| $U_6$ | | | | | | | | | | | | | |
| $U_7$ | | | | | | | | | | | | | |
| $U_8$ | | | | | | | | | | | | | |

The three-dimensional encoding is explained as follows:

The optical bandwidth is divided into four wavelengths, $Y_1$, $Y_2$, $Y_3$, and $Y_4$, according to the spectral sequence $X_1$ = [1100] and $X_2$ = [0011] ([$Y_1Y_2Y_3Y_4$]), where the following four wavelengths are: $Y_1$ = 1550 nm, $Y_2$ = 1552 nm, $Y_3$ = 1554 nm, and $Y_4$ = 1555 nm, as four optical bandwidths between (1550–1555 nm). Similarly, for the time coding, it is divided into four different time delays: $\tau_1$ = 0 ns, $\tau_2$ = 0.25 ns, $\tau_3$ = 0.5 ns, and $\tau_4$ = 0.75 ns, according to the time sequence $Y_1$ = [1100] and $Y_2$ = [0011] ([$\tau_1\tau_2\tau_3\tau_4$]). Moreover, the spatial coding is also divided in the same way, into four branches of couplers $C_1$, $C_2$, $C_3$, and $C_4$, according to the spatial sequence $Z_1$ = [1100] and $Z_2$ = [0011]([ $C_1C_2C_3C_4$]); consequently, we connect each component with the other according to the 3D-VWZCC code.

In the first part, each user contains an information source (pseudo-random bit sequence (PRBS), white light-type optical generator (white light source), non-return-to-zero pulse generator (NRZ Pulse Generator), electrical/optical modulator (Mach-Zehnder Modulator MZM), two sets of fiber Bragg gratings FBG1 and FBG2, two timers, and star couplers.

Firstly, the PRBS generator generates a bit stream of 1Gbps and converts this data to an NRZ line code signal (−1, +1), then, an optical source is sent to the spectral encoder FBG1 to select the desired wavelengths; afterwards, the selected wavelengths are reflected and sent to the FBG2 encoder for compensation for the run-trip delay according to the spectral sequence. Then, the optical code is modulated with the NRZ signal and converted to optical domain using an external MZM modulator, secondly, the coded pulses are sent to the delay line for time encoding, and thirdly, the delay line output is connected with star couplers for spatial encoding. In this case, the signal is encoded to 3D-VWZCC.

In the second part, each receiver contains star couplers, two timers, two sets of fiber Bragg gratings FBG1 and FBG2, PIN photo-detector, and low pass filter (low pass Bessel filter LPF).

Firstly, the received signal is divided and sent to the delay line for the temporal decoding according to the temporal sequence, then to the spectral decoder FBG for the spectral decoding according to the spectral sequence, secondly, the decoded impulses are sent to the photo-detector for conversion into electrical signals, and finally, the output electrical signals are sent to the low pass Bessel filter in order to filter the unwanted signals and keep useful signals.

Figure 16 shows the opening of the eye diagrams of eight users where the data rate equals 1 Gbps; according to the figure, we note that the eight users have a maximum vertical opening, which indicates that the proposed code offers better performance and that the data transmitted are received almost as the data that are sent. Our code offers a quality

factor Q, which reaches between 9 and 11.56 with a BER reaching between $7.85 \times 10^{-20}$ and $1.99 \times 10^{-31}$ at a high data rate. To strengthen and prove these results, Figures 17 and 18 show the eye diagram of one of users at a data rate of 1 Gbps and 2 Gbps, respectively. Moreover, the user providing 1 Gbps of data rate shows a quality factor reaching 11.56 and corresponds to a BER of $1.99 \times 10^{-31}$, whereas the user who provides 2 Gbps of data rate can reach a quality factor of 9.45 with a BER equal to $7.96 \times 10^{-22}$; hence, we explain these results by the high efficiency offered by our code at a high data rate despite the existence of a high number of users, due to low impact of MAI effect and PIIN noise, and this is thanks to the zero cross-correlation property of our code.

In addition to that, the primary principle of the interpretation of the eye diagram is to evaluate the aperture regardless of whether it is vertical or horizontal in terms of BER and quality factor Q; in addition to that, other parameters are also considered, such as jitter, noise margin, and time sensitivity. For that, from Figures 17 and 18 we evaluate the following parameters: firstly, in terms of jitter, we notice that the jitter of the eye diagram of our code is very low. Secondly, in terms of noise margin, the eye-opening is at the maximum, which indicates that the signal-to-noise ratio SNR is very high and the noise power is very low; this due to the ZCC property of 3D-VWZCC code, which suppresses PIIN noise. Finally, in terms of time sensitivity, the time-sensitivity slope of the eye is small, which indicates that the sensitivity to synchronization error is very low. In summary, the proposed 3D-VWZCC code demonstrates a high immunity against transmission errors due to zero cross-correlation properties, which totally ignores multiple access interference (MAI) and reduces PIIN noise.

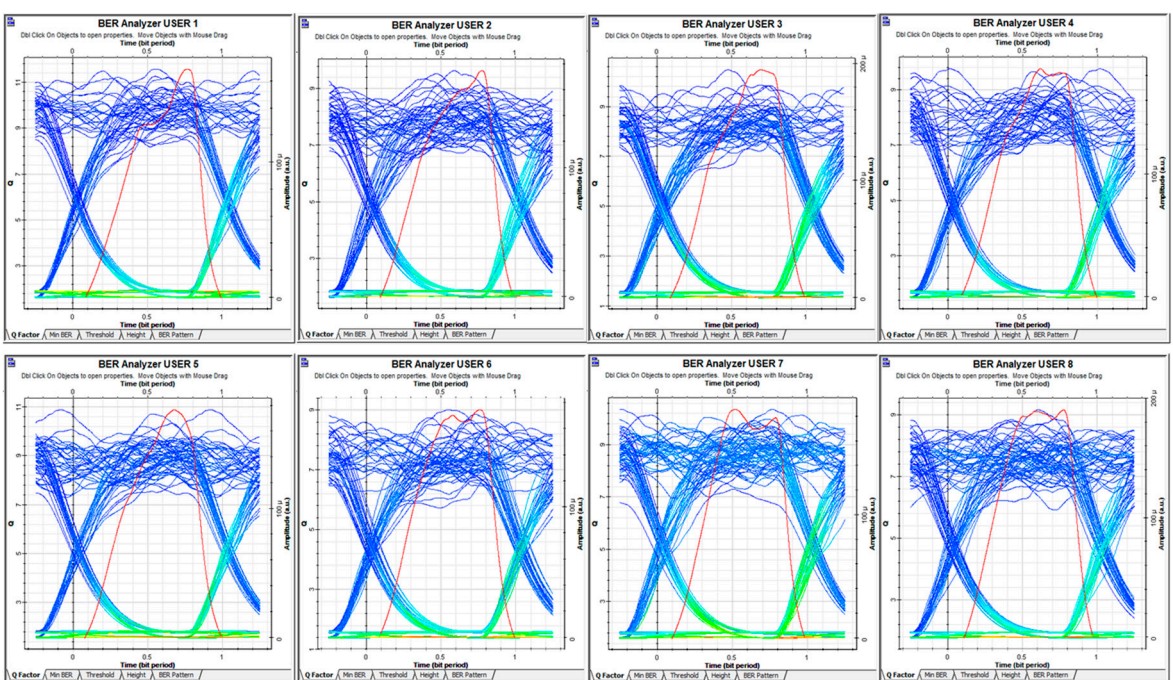

**Figure 16.** Eye diagrams of eight 3D-VWZCC users.

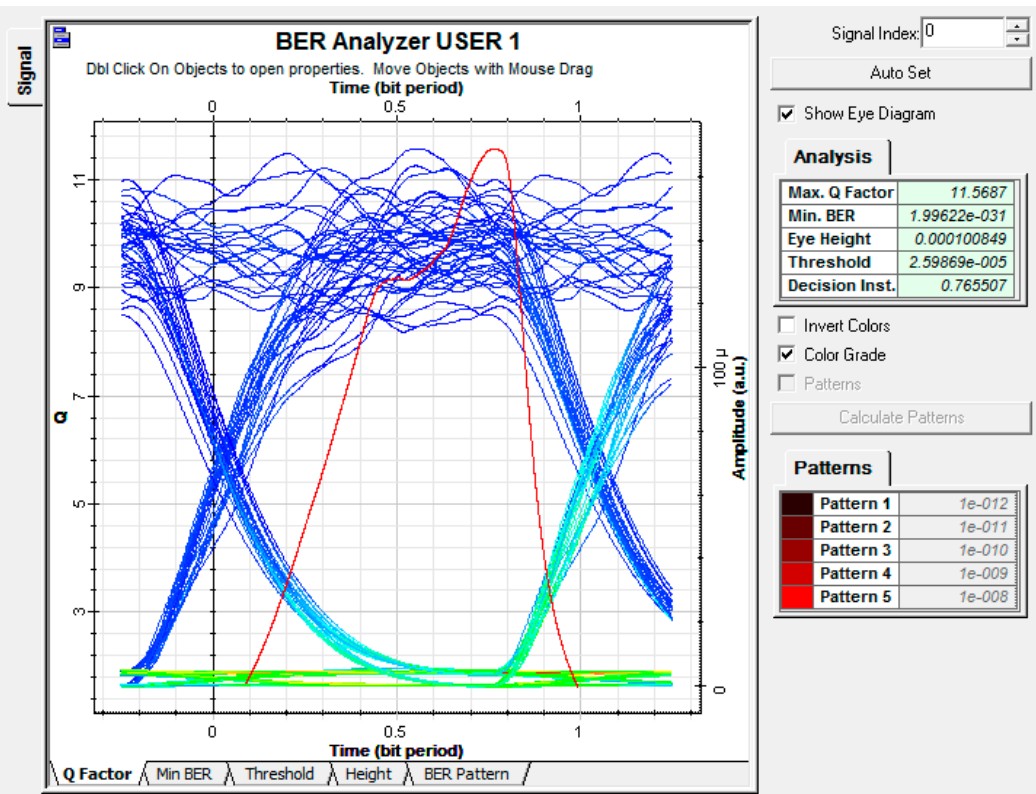

**Figure 17.** Diagram of the first user's eye at 1 Gbps.

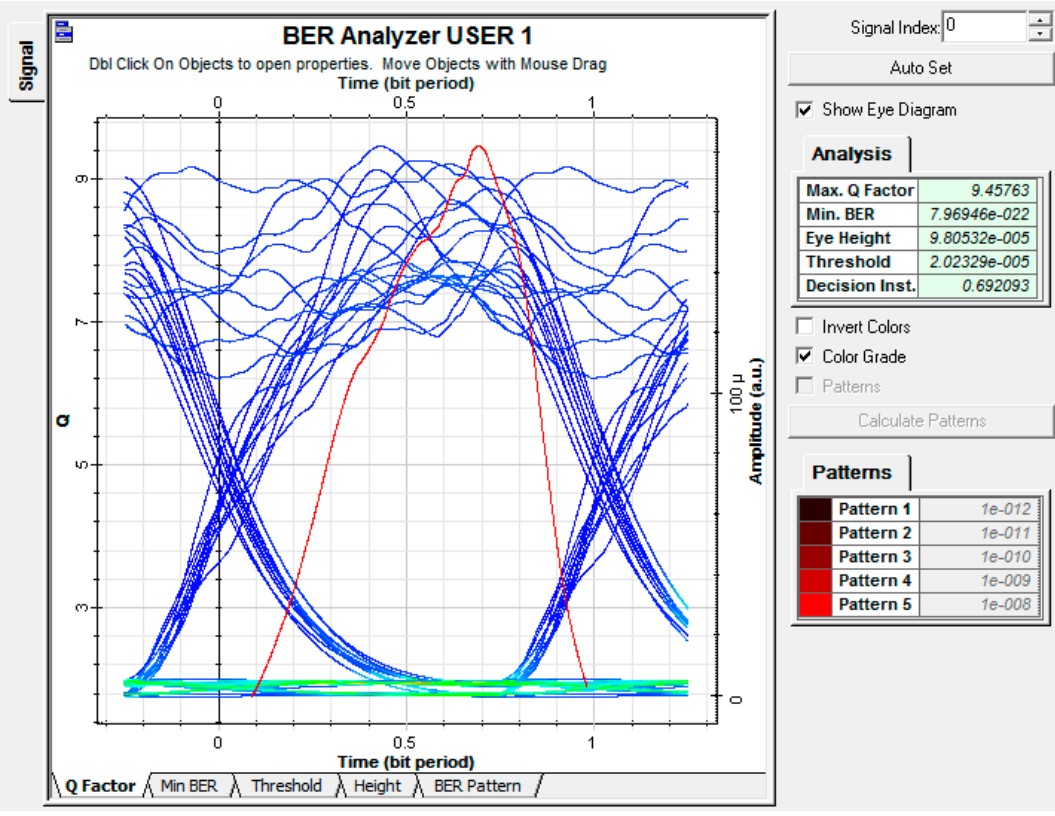

**Figure 18.** Diagram of the first user's eye at 2 Gbps.

## 8. Conclusions

In this study, a new three-dimensional code for spectral/time/spatial was suggested based on the one-dimensional VWZCC code extension. Moreover, the proposed code is characterized by the property of zero cross-correlation (ZCC), high immunity against PIIN noise and multiple access interference (MAI), supports a large number of simultaneous users, supports a very high data rate, and reduces system complexity due to the use of a single photodetector at the receiver level. The performance of our system demonstrates that the suggested code performs well compared to 3D-PD, 3D-PD/MD, and 3D-DCS/MD codes. Further, the comparison of this code from the 1D and 2D to 3D passage proved a better performance in terms of BER and SNR as well as a quality factor Q, which makes the reception of a received signal power higher than the noise power. According to the numerical results, the proposed code improves the optical system capacity up to 3.686, 3.234, 2.908, 5.112, and 2.248 times compared to 3D-PD, 3D-PD/MD, 3D-DCS/MD, 2D-VWZCC, and 1D-VWZCC codes. It saves a received power up to $-7.04$ dBm and $-5.9$ dBm compared to the 1D-VWZCC and 2D-VWZCC codes, respectively. Furthermore, the suggested system is successfully implemented using Optisystem software, where the code provides a low BER up to $1.99 \times 10^{-31}$ and a better quality reach, up to 11.56. The proposed code demonstrates high efficiency due to the absence of MAI and the low effect of PIIN noise compared to previously published codes, which allows the system to meet the requirements of high-speed optical transmission networks with a simple transmitter/receiver structure and supports a large number of users at a very high data rate and low power consumption with a higher quality factor Q (greater than 6) and a very low bit error rate (less than $10^{-9}$), which makes it executable in future optical networks, such as the Wavelength Division Multiplexing (WDM), Next-Generation Passive Optical Network (NG-PON), free-space optics (FSO), etc.

**Author Contributions:** Conceptualization, M.R.; methodology, M.R. and A.C.; software, M.R. and A.C.; validation, A.C., A.S.K. and G.N.S.; formal analysis, M.R.; investigation, A.C., A.S.K., B.S.B. and G.N.S.; resources, A.S.K.; data curation, M.R. and A.C.; writing—original draft preparation, M.R.; writing—review and editing, A.C., A.S.K., B.S.B. and G.N.S.; visualization, A.C. and G.N.S.; supervision, G.N.S. and A.C.; project administration, A.C. and G.N.S.; funding acquisition, A.S.K. All authors have read and agreed to the published version of the manuscript.

**Funding:** This research received no external funding.

**Institutional Review Board Statement:** Not applicable.

**Informed Consent Statement:** Not applicable.

**Data Availability Statement:** Not applicable.

**Acknowledgments:** This research is supported by the General Directorate of Scientific Research and Technological Development 'DGRST'—Algeria.

**Conflicts of Interest:** The authors declare no conflict of interest.

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
