# Peer review of "Contribution of New Three-Dimensional Code Based on the VWZCC Code Extension in Eliminating Multiple Access Interference in Optical CDMA Networks"

_photonics, doi:10.3390/photonics9050310_

Round 1

Reviewer 1 Report

  1. The literature review is insufficient. A detailed literature review with a table should be added.
  2.  The simulation study is inconclusive. The results presented are vague and not conclusive. 
  3. The eye diagrams do not convey any information. What does it convey? What are the eye-opening penalties?
  4. Table 5 is not complete. All the parameters should be added. 
  5. The quality of the work is very low. In addition to SNR and BER, EVM results should be added too in order to see how constellations and Eye diagrams correlate. 

Author Response

Dear Reviewer

Please find attached our answer to your questions + our corrected manuscript

we would like to thank the reviewers for the appreciation given to our work, also for their valuable comments and relevant suggestions. We have provided detailed answers to their comments and mentioned the changes brought according to their suggestions. The reviewers’ comments are in black font and the answers are in blue font, revisions on the manuscript are in green. We hope we have correctly addressed the questions recommended by the reviewers. 

Reviewer 2 Report

The paper is well written and includes detailed explanations and lots of results.

Several drawbacks though, as follows: 

  •  the pool of referenced papers is kind of old (out of 29 references only 7 have been published during the last 2 years). The authors should make a fresh literature search and should find papers with similar topics as references.
  • Figure 1 is not referred in the text and not explained. It has been deduced by the authors or taken from existing literature (in the former case, citation is necessary)
  • the results presented in Figures 3-12 are compared with similar results from papers published in 2009-2019 [13-15]. The authors may refer also to newer findings to check if their approach is still actual. Also, the size of the figures is too large.
  • Figure 13 has a very poor resolution, and irrelevant. We believe the authors when saying that the simulation was performed in Matlab it is not necessary to put the diagram.
  • Table 6 is not explained; do the colours from that table have any particular significance?
  • Figure 14 is made up of 8 sub-figures, but the sub-figures are not identified in any way and are not discussed in the text. What is the role of Figure 14 there then? 
  • The eye patterns should be exploited more thoroughly - evaluate the jitter, the noise margin, time sensitivity and so on. Just showing them without using them is not enough. 
  • The conclusion section should be extended to include all the findings of the present work as well as future research opportunities. 

Author Response

Dear Reviewer
Please find attached our answer to your questions + our corrected manuscript.

we would like to thank the reviewers for the appreciation given to our work, also for their valuable comments and relevant suggestions. We have provided detailed answers to their comments and mentioned the changes brought according to their suggestions. The reviewers’ comments are in black font and the answers are in blue font, revisions on the manuscript are in green. We hope we have correctly addressed the questions recommended by the reviewers. 

Reviewer 3 Report

Authors preseny new three-dimensional spectral / time / spatial Variable Weight Zero Cross Correlation code for non-coherent Spectral Amplitude Coding Optical Code Division Multiple Access. Authors also presenting simulation results show that their code proves high immunity against PIIN noise and shot noise, and also increases multiplexing ability. Presented code offers also better performance in terms of data rate up to 2 Gbps compared to previous codes which makes the system meet the requirements of optical communication networks.

The article was written correctly and it raises a very interesting problem. The authors presented the obtained results in a very extensive way, together with their explanation and analysis. The obtained results clearly show the advantages of the proposed code. 

I have only a few small remarks that do not detract from the significance of the presented results:

  • Figure 3: legend should be in the bottom right. In the present positon it ovelaps with lines.
  • Figure 5: Spectral Width axis: unit should be "THz" not "Thz".
  • Figure 7: legend should be in the bottom right. In the present positon it ovelaps with lines.
  • Figure 9 and 10: legend should be in the top left. In the present positon it ovelaps with lines.

Author Response

Dear Reviewer
Please find attached our answer to your questions + our corrected manuscript.

we would like to thank the reviewers for the appreciation given to our work, also for their valuable comments and relevant suggestions. We have provided detailed answers to their comments and mentioned the changes brought according to their suggestions. The reviewers’ comments are in black font and the answers are in blue font, revisions on the manuscript are in green. We hope we have correctly addressed the questions recommended by the reviewers

Round 2

Reviewer 1 Report

In Table 6, the Centre frequency is in nm, why? Shouldn't this be Wavelength?

What modulation type and rates were used?

In terms of EVM, the performance % is too high, the rate of EVM w.r.t 3GPP is 3.5 for 256 QAM and 8% for 64 QAM, etc. Why is it so high? Please justify.

Redraw Fig. 2, it is still not clear.

SNR, BER and EVM are too high. This doesn't convinces me with the facts provided by the authors. The results  should be in limits provided by the standardization work items such as 3GPP. 

EOP does convey somthing! in the case of high non linearities, BER and eye openings are degraded. The EOP is not clean as expected if it had been a good match. Please provide Q factor results to justify the results. 

Author Response

Dear Editor and Reviewers,

    First, we would like to thank the reviewers for the appreciation given to our work, also for their valuable comments and relevant suggestions. We have provided detailed answers to their comments and mentioned the changes brought according to their suggestions. The reviewers’ comments are in black font and the answers are in blue font, revisions on the manuscript are in green. We hope we have correctly addressed the questions recommended by the reviewers.

Reviewer 2 Report

The paper has significantly improved now and I think it can be published in the present form.

Round 3

Reviewer 1 Report

Accepted in present form. The clarifications presented on the queries regarding modulation and EVM has been clarified. I am still doubtful on Eye diagrams but I will give an acceptance given that it is an ongoing research work.